# Analysing the mechanism of mitochondrial oxidation-induced cell death using a multifunctional iridium(III) photosensitiser

Chaiheon Lee [1,2,5], Jung Seung Nam [1,2,5], Chae Gyu Lee [1,2], Mingyu Park[1,2], Chang-Mo Yoo[3], Hyun-Woo Rhee [3], Jeong Kon Seo[4 ✉] & Tae-Hyuk Kwon [1,2 ✉]

Mitochondrial oxidation-induced cell death, a physiological process triggered by various cancer therapeutics to induce oxidative stress on tumours, has been challenging to investigate owing to the difficulties in generating mitochondria-specific oxidative stress and monitoring mitochondrial responses simultaneously. Accordingly, to the best of our knowledge, the relationship between mitochondrial protein oxidation via oxidative stress and the subsequent cell death-related biological phenomena has not been defined. Here, we developed a multifunctional iridium(III) photosensitiser, Ir-OA, capable of inducing substantial mitochondrial oxidative stress and monitoring the corresponding change in viscosity, polarity, and morphology. Photoactivation of Ir-OA triggers chemical modifications in mitochondrial protein-crosslinking and oxidation (i.e., oxidative phosphorylation complexes and channel and translocase proteins), leading to microenvironment changes, such as increased microviscosity and depolarisation. These changes are strongly related to cell death by inducing mitochondrial swelling with excessive fission and fusion. We suggest a potential mechanism from mitochondrial oxidative stress to cell death based on proteomic analyses and phenomenological observations.

[1] Department of Chemistry, Ulsan National Institute of Science and Technology (UNIST), Ulsan 44919, Republic of Korea. [2] Center for Wave Energy Materials, Ulsan National Institute of Science and Technology (UNIST), Ulsan 44919, Republic of Korea. [3] Department of Chemistry, Seoul National University, Seoul 08826, Republic of Korea. [4] UNIST Central Research Facility, Ulsan National Institute of Science and Technology (UNIST), Ulsan 44919, Republic of Korea. [5] These authors contributed equally: Chaiheon Lee, Jung Seung Nam. ✉email: jkseo6998@unist.ac.kr; kwon90@unist.ac.kr

Elucidating the process of mitochondrial oxidation-induced cell death is essential to understanding and improving cancer therapeutics based on oxidative damage to tumours[1–4]. To this end, various mitochondria-targeted photosensitisers that can produce reactive oxygen species (ROS) have been utilised to understand the underlying chemical effect of mitochondrial oxidation[5,6].

Oxidative stress induced by photosensitisers causes chemical modifications of biomolecules including proteins[7–9], unsaturated lipids[10,11], and DNA[12,13]. Notably, protein modifications occurring via methionine oxidation and dityrosine crosslinking are clearly described chemical modifications for mitochondrial oxidation-induced cell death[9,14]. However, the connection between the chemical modifications of mitochondrial proteins and the biological response in cell death remains elusive due to the absence of a chemical tool to analyse the biological phenomena (i.e., environmental changes in mitochondrial surroundings in terms of viscosity, polarity, morphology, pH, and temperature) that occur in mitochondria in response to oxidative stress. Therefore, photosensitisers that efficiently oxidise proteins and subsequently monitor the consequent protein dysfunction-related mitochondrial responses are needed[15–17]. To address this, we employed organometallic iridium(III) complexes because of their notable ROS generation efficiency, lifetime sensitivity to microviscosity, straightforward ligand tuning, cell permeability, and photostability[16,18,19].

In this work, our designed iridium(III) complex, Ir-OA, showed accelerated ROS production and micropolarity-dependent ratiometric emission properties owing to intramolecular energy transfer system. With these characteristics, we successfully employed Ir-OA to (i) induce cell death by producing ROS in the mitochondria, (ii) monitor changes in the mitochondrial microenvironment (i.e., viscosity, polarity, and morphology) using various techniques including lifetime and ratiometric imaging, and (iii) identify the mode of action for microenvironment changes by profiling the modified proteins (i.e., crosslinked and oxidised proteins) through oxidative stress. Collectively, we suggest a promising mechanism that describes how oxidative stress affects the mitochondria and induces cell death, corroborating the correlation between mitochondrial microenvironment changes and the oxidised proteome. The proposed mechanism may aid the understanding of how some cancer therapeutics can induce mitochondrial oxidative stress.

## Results

**Characterisation of iridium(III) complexes**. We synthesised iridium(III) complexes with (Ir-OA) and without (Ir-OC) energy donors (Fig. 1a, b, and Supplementary Figs. 1–9). Previous studies have revealed that cationic iridium(III) complexes with 2-phenylquinoline (2pq) ligands effectively generated a singlet oxygen ($^1O_2$) and superoxide radical anion ($O_2^{•-}$) owing to their suitable HOMO and LUMO levels for energy and electron transfers to molecular oxygen[9]. Nevertheless, the iridium(III) complex with 2pq ligands displayed an insufficient absorption coefficient ($<6000\ M^{-1}cm^{-1}$) in the 400–500 nm range, resulting in a limited ROS production (Supplementary Table 1). Accordingly, we introduced an acedan (6-acetyl-2-(dimethylamino) naphthalene)-based energy donor to the iridium(III) photosensitiser (Ir-OA). Acedan derivatives have a strong absorption coefficient at 400 nm ($>16,000\ M^{-1}cm^{-1}$) and high emission quantum yields ($>0.90$) in the range of 400–500 nm (Supplementary Table 1)[20]. The emission spectrum of energy donors and the absorption spectrum of the iridium(III) complex with 2pq ligands overlapped well (Supplementary Fig. 10), allowing efficient intramolecular energy transfer (Fig. 1a). The energy donor

and acceptor of Ir-OA absorbed light independently due to its non-conjugated bridge, resulting in the absorption spectrum of Ir-OA corresponding to the sum of the absorption spectra of Ir-OC and compound 4 (Fig. 1c). In addition, the fluorescence of the energy donor at 470 nm was nearly quenched in Ir-OA ($\lambda_{ex}$ = 385 nm, the absorption peak of the energy donor). Simultaneously, the emission of the energy acceptor at 555 nm was enhanced, which strongly suggests that highly efficient intramolecular energy transfer occurred ($\varphi_{energy\ transfer}$ = 98.4%) (Fig. 1d).

**In vitro/live cell ROS generation**. To verify the capacity of Ir-OA and Ir-OC to generate ROS, we conducted an $^1O_2$ generation assay using 9,10-anthracenediyl-bis(methylene) dimalonic acid (ABDA), and an $O_2^{•-}$ assay using dihydrorhodamine123 (DHR123)[21]. Ir-OA in the ABDA solution showed greater $^1O_2$ generation compared with Ir-OC, compound 4, and $[Ru(bpy)_3]^{2+}$ as a reference photosensitiser (Fig. 1e). Moreover, Ir-OA showed much stronger $O_2^{•-}$ generation than Ir-OC and compound 4 in the DHR123 assay (Fig. 1f)[22]. This implies that more triplet excitons were produced by intramolecular energy transfer of Ir-OA, which accelerated $^1O_2$ and $O_2^{•-}$ generation. The ROS indicator 2′,7′-Dichlorodihydrofluorescein diacetate (H$_2$DCF-DA) was utilised to confirm whether ROS were generated inside live cells by the photoactivation of Ir-OA (Supplementary Fig. 11)[23]. When HeLa cells were treated with Ir-OA and H$_2$DCF-DA, photoactivation of Ir-OA under very weak light irradiation (LED array, $\lambda$ = 400 nm, 0.17 J cm$^{-2}$) produced ROS efficiently, resulting in a strong green signal. However, Ir-OC did not effectively induce an ROS signal under the same conditions as the light energy was not sufficient for Ir-OC to generate adequate ROS levels. According to the intracellular ROS assay (Supplementary Fig. 11), Ir-OA can be employed as an effective photosensitiser to induce strong oxidative stress to live cells and is expected to induce physiological dysfunction resulting in cell death.

**Localisation of iridium(III) complexes**. Ir-OA and Ir-OC contain cationic iridium cores and lipophilic ligands, which target mitochondria due to the negative mitochondrial membrane potential (MMP) at the inner mitochondrial membrane (IMM)[24]. We determined the subcellular location of Ir-OA and Ir-OC by confocal laser scanning microscopy (CLSM) using MitoTracker Deep Red, a fluorescent probe for IMM and intermembrane space (IMS) staining[25]. Noticeably, the CLSM images overlapped with those of the MitoTracker (Pearson's coefficient = 0.86 for Ir-OA and 0.74 for Ir-OC), which clearly indicated that Ir-OA and Ir-OC were located on the mitochondria (Fig. 2a and Supplementary Fig. 12). To identify the detailed location of Ir-OA in the mitochondrial ultrastructure, we conducted Airyscan confocal imaging ($\sim$120 nm)[26]. The Airyscan image of Ir-OA also overlapped with that of the MitoTracker and showed a clear boundary of the mitochondria corresponding to the IMM; this image also showed localisation of Ir-OA in the outer part of the IMM (Fig. 2b). To further distinguish the sub-mitochondrial localisation of Ir-OA, mitochondrial matrix targeted Mito-EGFP was transfected for matrix imaging with Ir-OA (Fig. 2c). The Airyscan image showed that Ir-OA was mainly located at the outer boundary of the Mito-EGFP signals, implying that the major population of luminescent Ir-OA is not in the mitochondrial matrix. This super-resolution imaging result suggests that the sub-mitochondrial localisation of Ir-OA might be from the IMM spreading to the IMS and outer mitochondrial membrane (OMM). Therefore, Ir-OA is expected to exert oxidative stress not only on mitochondrial proteins but also on proteins of subcellular organelles, such as the endoplasmic reticulum (ER)[27] and peroxisomes[28,29], in contact with the

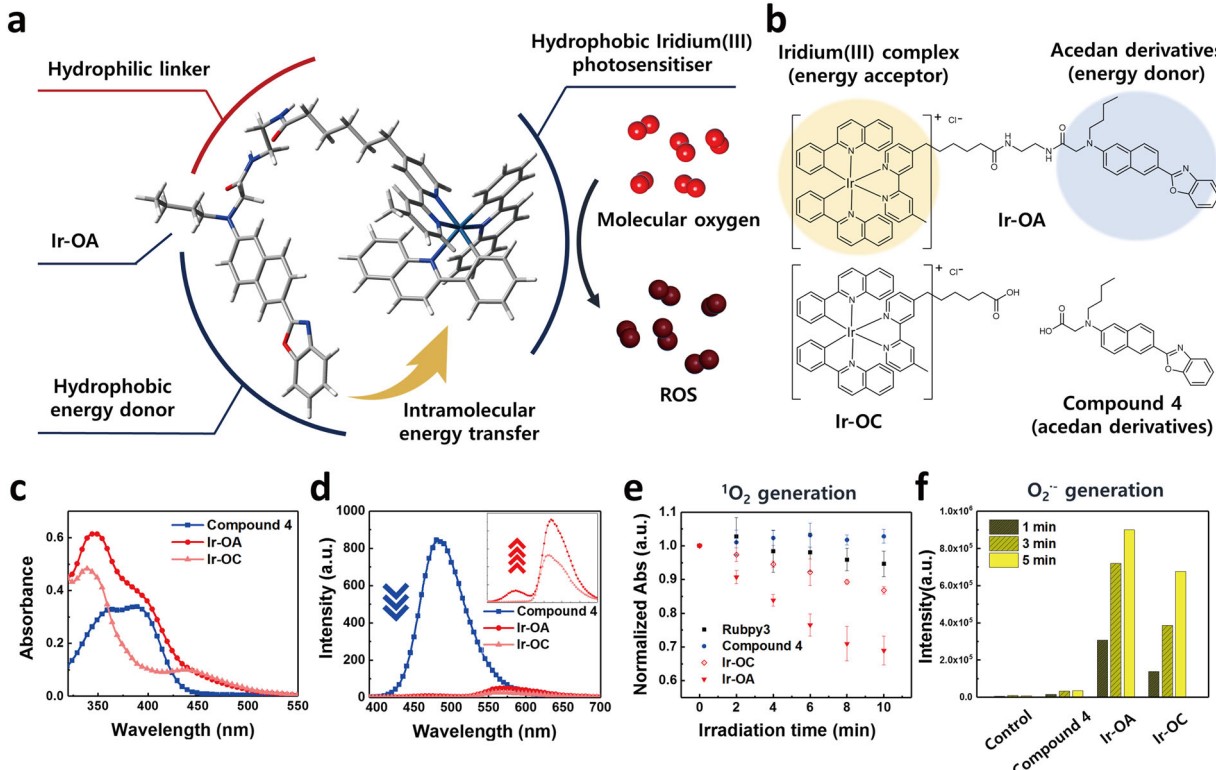

**Fig. 1 Photophysical characterisation of iridium(III) photosensitiser. a** Schematic illustration of the molecular engineering strategy: intramolecular energy transfer and resulting photoactivated ROS generation. The illustrated Ir-OA molecule is a Δ isomer, but the Λ form enantiomer can exist. **b** Molecular structure of Ir-OA, Ir-OC, and compound 4. **c** UV-vis absorption spectrum of the three presented chemicals. **d** Subsequent emission spectra ($\lambda_{ex} = 400$ nm) of the three chemicals. Magnified emission spectra show the enhanced emission of Ir-OA compared with that of Ir-OC, which reveals the evidence of energy transfer. Conditions for the absorption and emission spectra; [Ir-OA or Ir-OC or compound 4] = 20 μM in $H_2O$:DMSO = 99:1 (v/v%). **e** $^1O_2$ assay using the absorbance decay of 9,10-anthracenediyl-bis(methylene)dimalonic acid (ABDA) under light exposure. The ABDA is degraded by $^1O_2$ produced by photoactivation of the Ir-OA. Data are presented as mean value ± s. d. (n = 3). **f** $O_2^{•-}$ assay using the fluorescence enhancement of dihydrorhodamine 123 (DHR123). DHR123 is oxidised to rhodamine123 by the produced $O_2^{•-}$, which enhances fluorescence signal. Conditions for ROS assays: [iridium(III) complexes] = 4 μM; [ABDA] = 100 μM or [DHR123] = 4 μM in $H_2O$:DMSO = 999:1 (v/v%). Source data are provided as a Source Data file.

mitochondria, which may cause protein dysfunction, micro-environment changes, and cell death.

**Mitochondrial oxidation-induced cell death.** To confirm the resulting cell death by photoactivation of Ir-OA in mitochondria, we conducted the CCK-8 and MTT assays for quantitative cytotoxicity analyses and the live/dead assay with propidium iodide (PI) (dead cell indicator) and Calcein AM (live cell indicator)[30]. Cell viability dramatically decreased following photoactivation of Ir-OA under low light energy (0.08–0.25 J cm$^{-2}$) (Fig. 3a, Supplementary Fig. 13, and Supplementary Table 2). However, Ir-OC was unable to cause significant changes in cell viability as the generated ROS levels were insufficient to trigger cell death under the same conditions. Notably, the IC$_{50}$ value of Ir-OA obtained using the CCK-8 assay was considerably higher than that obtained using the MTT assay (Supplementary Table 2). This is because the formed formazan in the MTT assay depends on the mitochondrial dehydrogenase, while CCK-8 is activated by dehydrogenases from whole cells. Thus, the CCK-8 assay is more appropriate and reliable in measuring the phototoxicity of pho-tosensitisers targeting the mitochondria. Furthermore, the pho-totoxicity results of Ir-OA from the CCK-8 and MTT assays corresponded to results obtained from the live/dead assay. Intense PI fluorescence for dead cells was shown in HeLa cells with Ir-OA, whereas HeLa cells with Ir-OC exhibited the green

fluorescence of Calcein AM for live cells (Fig. 3b). In addition, cells with Ir-OA showed an enhanced PI signal within 90 min of light irradiation (LED array, λ = 400 nm, 0.17 J cm$^{-2}$), implying that cell death was initiated in less than 2 h (Supplementary Fig. 14), possibly due to the strong oxidation of mitochondria. The live/dead assay was confirmed by fluorescence activated cell sorting (FACS) and assessed using 2D histograms (Fig. 3c and Supplementary Fig. 15). When Ir-OA was photoactivated, the number of HeLa cells with positive PI and negative Calcein AM dramatically increased, confirming that most of the cells were dead. In addition, the number of cells with positive Annexin V and negative PI (Q2) increased upon photoactivation with Ir-OA prior to the disruption of the cell membrane (positive PI) (Fig. 3d and Supplementary Fig. 15), showing that the dying cells undergo an early apoptosis stage. Hence, the photoactivation of Ir-OA triggered apoptotic cell death while that of Ir-OC did not alter cell viability.

**Mitochondrial viscosity change by protein-crosslinking.** ROS and following oxidative stress triggers a change in micro-environment such as viscosity[16,31] and polarity[15,32] at the specific region affected by the stress. Particularly, increased viscosity affects biomolecular interactions and metabolite diffusion, which hinders mitochondrial respiration and metabolism[31,33] and, subsequently, induces cell death. Although protein crosslinking

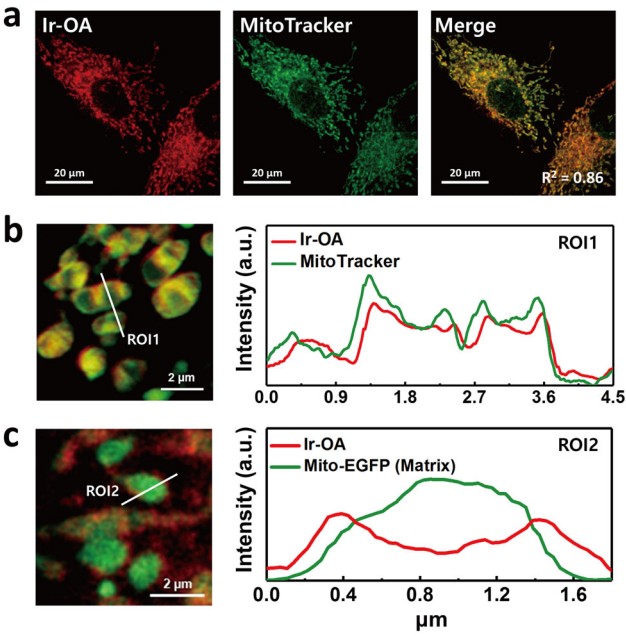

**Fig. 2 Localisation of Ir-OA in living cells. a** Confocal images of Ir-OA with MitoTracker. Phosphorescence of Ir-OA (red) and fluorescence of MitoTracker (green) and merged image ($\lambda_{ex} = 405$ nm for Ir-OA, $\lambda_{ex} = 647$ nm for MitoTracker® Deep Red FM). Pearson's coefficient was calculated using Image J software ($R^2 = 0.86$). **b** Airyscan confocal image of mitochondria with Ir-OA (red) and MitoTracker (green, outer/inner mitochondria membrane) for identifying the specific location of Ir-OA. **c** Airyscan confocal image of mitochondria with Ir-OA (red) and Mito-EGFP (green, mitochondrial matrix). Line profiling for clarifying the specific location of each emission signal was followed by corresponding images (right). The Ir-OA signal is well merged with the MitoTracker signal on the outer/inner mitochondrial membrane and encloses Mito-EGFP signal of the mitochondrial matrix. The mitochondria of Airyscan images were pre-swelled by photoactivation ($\lambda_{ex} = 405$ nm, 0.0125 mW) of the Ir-OA for identifying mitochondrial substructure (the mitochondrial swelling effect of the Ir-OA is explained later). Line profiling analysis was proceeded with Carl Zeiss ZEN 3.0 Blue software. Conditions: [Ir-OA] = 4 μM, [MitoTracker® Deep Red] = 100 nM, incubation time = 2 h and 0.5 h, respectively. All imaging was repeated three times independently, and each experiment showed similar results. Source data are provided as a Source Data file.

has been proposed as a possible cause of viscosity change, no experimental evidence has been previously reported.

Generally, ROS are known to crosslink proteins and generate protein aggregates[9,34], which can increase the viscosity of the microenvironment. The viscosity change can affect the lifetime of iridium(III) complex; lifetime change was measured by time correlated single photon counting (TCSPC). Firstly, the viscosity sensitivity of the Ir-OA phosphorescence lifetime was measured in MeOH and glycerol. The phosphorescence lifetime of Ir-OA increased from 278 ns to 2293 ns as the glycerol content (v/v) increased from 0% (0.55 cP) to 95% (950 cP) (Fig. 4a and Supplementary Fig. 16). In addition, we confirmed that the increased protein concentration at the local area resulted in the phosphorescence lifetime enhancement of Ir-OA. Bovine serum albumin (BSA), known to increase the viscosity of solutions according to its concentration[35], was dissolved in aqueous solutions of Ir-OA at the following concentrations: 0.0156, 0.0625, 0.250, 1.00, and 4.00 mg/mL (Supplementary Fig. 17). Further, the phosphorescence lifetime of Ir-OA was measured and showed a concentration-dependent increase from 664 ± 31 ns

to 1912 ± 40 ns. This result implies that the local accumulation of proteins by photocrosslinking, and the corresponding viscosity increase, can be monitored by the change in the phosphorescence lifetime of Ir-OA. Accordingly, in HeLa cells, we used phosphorescence lifetime imaging microscopy (PLIM) to monitor changes in mitochondrial viscosity as a product of oxidative stress caused by Ir-OA (Fig. 4b). HeLa cells with Ir-OA exhibited an average lifetime of 866 ± 20 ns before light irradiation (LED array, $\lambda = 400$ nm, 0.17 J cm$^{-2}$), which increased to 915 ± 14 ns following irradiation, likely due to the accumulation of crosslinked proteins. Then, we further investigated whether Ir-OA successfully induced protein crosslinking with light irradiation (LED array, $\lambda = 400$ nm, 1.28 J cm$^{-2}$) in live cells. In addition, the photocrosslinking reaction by Ir-OA was confirmed in each organelle. First, we transfected four different EGFP constructs (Mito-EGFP, mitochondrial matrix; Sec61b-EGFP, ER membrane; PEX16-EGFP, peroxisome; and PTBP1-EGFP, nucleus) (Supplementary Table 3) in the cells, followed by incubation of Ir-OA in the presence or absence of light. Western blot signals of covalently crosslinked EGFP were observed for Mito-EGFP, Sec61b-EGFP, and PEX16-EGFP in the presence of light (hv+) (Fig. 4c–f, left and Supplementary Fig. 18). However, the signal on a gel with anti-EGFP did not change after photoactivation of Ir-OA in the nucleus (PTBP1-EGFP) as the nucleus is further from the mitochondria and has no direct contact site.

Line-cut analysis comparing the presence or absence of light provided a correlation value ($R^2$), which was utilised to calculate crosslinking efficiency (η) (calculation details are explained in Supplementary Information) (Fig. 4c–f). Photocrosslinking was significantly generated in the ER membrane (η = 60.3%), mitochondrial matrix (η = 33.2%), and peroxisome (η = 20.6%) but not in the nucleus (η = 1.0%). The crosslinking efficiency difference is affected by the possibility of contact between Ir-OA and EGFP of each organelle. Therefore, we transfected four different EGFP constructs again and imaged the EGFP with Ir-OA to investigate their proximity (Fig. 4c–f, right). The Pearson's coefficients R of each image were then calculated (Mito-EGFP, R = 0.907; Sec61b-EGFP, R = 0.477; PEX15-EGFP, R = 0.124; PTBP1-EGFP, R = −0.347 vs. Ir-OA). Considering that the EGFP closer to Ir-OA is expected to be more easily crosslinked, it can be reasoned that the EGFP in the mitochondrial matrix, ER membrane, and peroxisome were more crosslinked than the EGFP in the nucleus. Notably, the crosslinking efficiency of ER membrane proteins was more significant than that of mitochondrial matrix proteins and peroxisome proteins because Ir-OA is located on the outer surface of the OMM and IMM. The ER membrane proteins were in direct contact with Ir-OA, while the contact between the proteins in the mitochondrial matrix and Ir-OA is limited by the IMM. Therefore, the Ir-OA molecules triggered relatively more protein crosslinking in the ER membrane than that in the mitochondrial matrix. In addition, the EGFP crosslinking efficiency was measured in HeLa cells; the tendency was similar to that of HEK293T cells (Supplementary Fig. 19). Protein crosslinking could be a possible way to increase viscosity in cells by inducing aggregation of mitochondrial or mitochondria-contacting organelle proteins (i.e., proteins in the mitochondrial matrix, ER membrane, and peroxisome). The protein crosslinking-induced viscous mitochondrial environment can affect diffusion-mediated cellular processes, such as mitochondrial metabolism, transport, and signalling—by reducing biomolecular diffusion and reaction rates. Consequently, the locally increased viscosity of mitochondria accelerates cell death.

**Mitochondrial depolarisation and related oxidised-proteome.**
Mitochondrial oxidative stress triggers mitochondrial depolarisation,

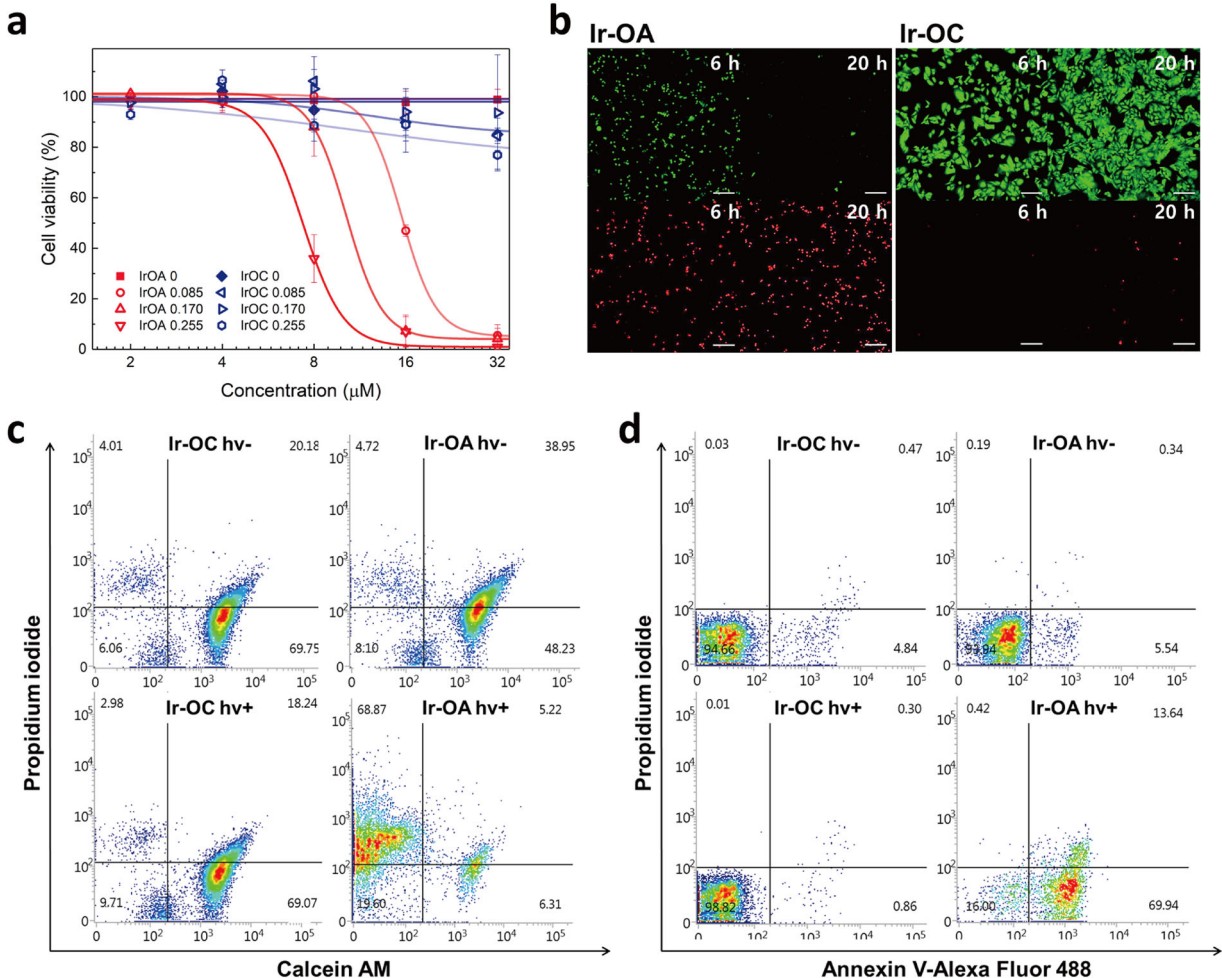

**Fig. 3 Identification of phototoxicity effect from iridium(III) complexes. a** CCK-8 assay for quantifying the cytotoxicity of Ir-OA and Ir-OC with or without light irradiation for HeLa cells. Data are presented as mean value ± s. d. ($n = 4$). Conditions: photosensitiser iridium(III) complexes incubation time = 2 h, light source = 400 nm light LED array, light dose: 0.085, 0.170, and 0.255 J cm$^{-2}$. **b** Live/dead assay for verifying phototoxicity of Ir-OA and Ir-OC. At 6 and 20 h after light irradiation, dead and live cells were stained using propidium iodide (PI, red) and Calcein AM (green), respectively (scale bars = 200 µm). Conditions: [Iridium(III) complex] = 8 µM, light source = 400 nm light LED array, light dose = 0.255 J cm$^{-2}$. The experiment was repeated three times independently, and each experiment showed similar results. **c** Representative Calcein AM vs. PI flow cytometry plot for HeLa cells incubated with iridium complexes. **d** Representative Annexin V vs PI flow cytometry plot for HeLa cells incubated with iridium complexes. Q1 (right top): late apoptotic/necrotic cells, Q2 (right bottom): early apoptotic cells, Q3 (left bottom): viable cells, and Q4 (left top): necrotic cells. Conditions for flow cytometry: [Ir complex] = 8 µM, light source = 400 nm light LED array, light dose = 0.25 J cm$^{-2}$. Source data are provided as a Source Data file.

which is the collapse of MMP. To monitor the change of mitochondrial depolarisation, intramolecular energy transfer was utilised because its efficiency is highly dependent on the surrounding polarity, thereby providing a ratiometric emission property (Fig. 5a). In pure MeOH solvent, the excited Ir-OA ($\lambda_{ex}$ = 400 nm, absorption maximum of donor) exhibited poor emission of acceptor ($\lambda_{em}$ = 555 nm) with strong emission of energy donor ($\lambda_{em}$ = 450 nm) due to its inefficient intramolecular energy transfer under relatively hydrophobic conditions (Fig. 5b). However, increasing water content caused the emission of acceptor ($\lambda_{em}$ = 555 nm) to become gradually enhanced while the donor emission ($\lambda_{em}$ = 450 nm) was reduced as efficient energy transfer occurred with increasing polarity. This is because the increasing hydrophilic environment forced the energy donor (acedan derivative) to move closer to the iridium ligands through π–π interactions, resulting in enhanced energy transfer efficiency (See Supplementary Information and Supplementary Fig. 20). We also confirmed the emission intensity profile inside cells according to polarity changes through lambda scanning (Supplementary Fig. 21).

By utilising this property, we represented ratiometric images monitoring mitochondrial depolarisation by oxidative stress (ratio = emission of acceptor, $\lambda_{em}$ = 573–620 nm/emission of donor, $\lambda_{em}$ = 420–480 nm) using CLSM (Fig. 5c). The mitochondria of HeLa cells with Ir-OA without inducing oxidative stress maintained high mitochondrial polarity (red, high MMP). Conversely, most mitochondria became gradually depolarised (blue, low MMP) within 90 min after photoactivation (0.17 J cm$^{-2}$) (Fig. 5c top). Furthermore, under continuous photoactivation, the emission ratio of Ir-OA exhibited an rapid change (green to blue) within 30 s in real-time analysis, implying rapid mitochondrial depolarisation by excessive oxidative stress (Fig. 5c bottom and Supplementary Movie 1). To support this, the conventional dye for detecting MMP loss, tetramethylrhodamine ethyl ester (TMRE), was utilised to confirm the time range of mitochondrial depolarisation. In HeLa cells with Ir-OA, the TMRE signal was completely quenched in 30 min after photoactivation (Supplementary Fig. 22). Consequently, we expect that proteins oxidised by a large amount of ROS are a primary cause of the strong and rapid depolarisation.

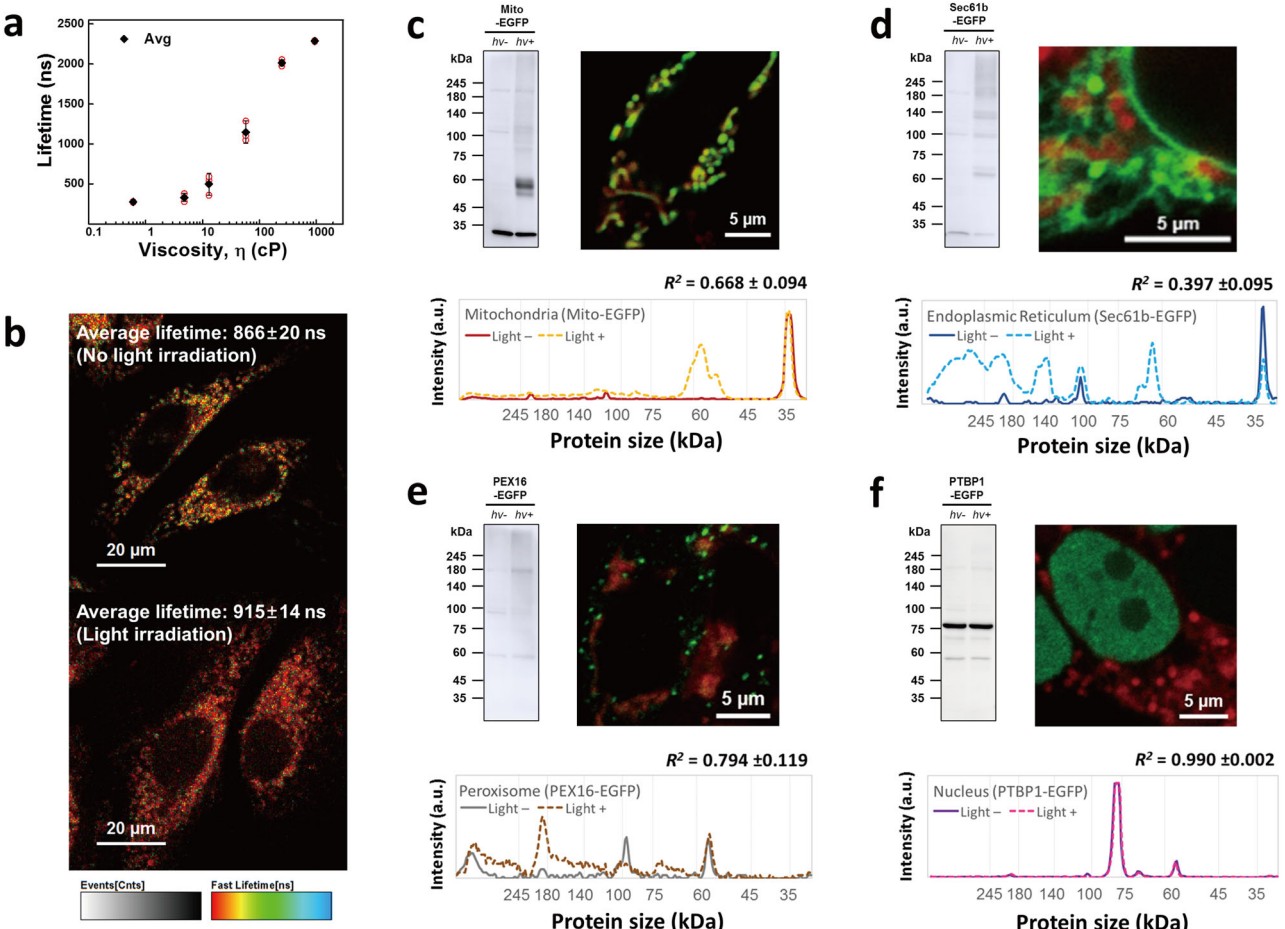

**Fig. 4 Mitochondrial viscosity changes with photo-crosslinking using photoactivation of Ir-OA. a** Viscosity-dependent change in the lifetime of Ir-OA. The in vitro viscosity was precisely controlled with glycerol content change in MeOH (v/v, %) from 0% to 95%. Data are presented as mean value ± s.d. ($n = 3$). **b** Phosphorescence lifetime image (PLIM) of Ir-OA in mitochondria before and after photoirradiation. The described lifetime is averaged value of three images obtained under same condition ($n = 3$). **c–f** Western blot (left) for identification of protein photo-crosslinking of Ir-OA depending on four different cell organelles: **c** mitochondria (Mito-EGFP); **d** ER (Sec61B-EGFP); **e** peroxisome (PEX16-EGFP); and **f** nucleus (PTBP1-EGFP). Co-localisation images (right) of Ir-OA (red signal) and each EGFP (green signal). $\lambda_{ex} = 405$ nm and 488 nm (Ir-OA and EGFP, respectively). Pearson's coefficient for respective cell organelles with Ir-OA was calculated using Image J software (Mito-EGFP, $R = 0.907$; Sec61b-EGFP, $R = 0.477$; PEX15-EGFP, $R = 0.124$; PTBP1-EGFP, $R = -0.347$ vs. Ir-OA). Line-cut analysis (bottom) of western blot signals with or without photo-irradiation was performed to quantify the crosslinking efficiency ($\eta$) ($n = 3$). Each correlation value ($R^2$) indicating similarity was written above the line cut spectrum. Note that higher efficiency for ER than mitochondria is for significant OMM/IMM location of Ir-OA. All imaging and blot experiments were repeated three times independently, and each experiment showed similar results. Source data are provided as a Source Data file.

To identify proteins that are significantly affected by depolarisation, the oxidised proteome in the whole cell was profiled because severe protein oxidation is one of the critical induction points for protein dysfunction[36,37]. As methionine is a common and easily oxidised amino acid, we analysed the oxidised-methionine (O-Met) in the whole cell proteome (Supplementary Data). First, proteomes with substantial O-Met were sorted by label-free quantification ($p$-value < 0.1) (Fig. 6a). Protein oxidation by photoactivated Ir-OA mainly occurred in the mitochondria, ER, and vesicles, and proteins in the cytoplasm and nucleus were partially oxidised as well, which closely corresponded to the crosslinking efficiency results (Fig. 6b). Next, we investigated 28 mitochondrial proteins among 112 substantially O-Met mitochondrial proteins. These were categorised according to their function: (i) channel and translocase (ii) oxidative phosphorylation (OXPHOS) complex. The proteins were visualised by heat map imaging (Fig. 6c). Among these, we focused on the oxidation of the VDAC1, VDAC3, and SLC25 family for the channel and translocase group, and ATP5A1 and

ATP5C1 for the OXPHOS complex group. The channel and translocase group is directly involved in MMP alteration because of mitochondrial cation exchange[38,39], especially that of H$^+$ and Ca$^{2+}$. VDAC1 and VDAC3 are indispensable OMM proteins that all ions and metabolites must cross before arriving at the mitochondrial inner space. Ions and metabolites that enter the inner mitochondrial membrane are subsequently transported to the matrix by the SLC25 family constituting mitochondrial carriers (MCs)[39,40]. Among the SLC25 family members, five proteins (SLC25A3, SLC25A5, SLC25A6, SLC25A10, and SLC25A24) were also significantly oxidised. Collectively, Ir-OA photoactivation substantially collapsed the MMP owing to severe oxidation of the two representative gatekeepers on the outer and inner mitochondrial membranes, thereby disturbing the movement of ions or transport metabolites.

Our O-Met proteome also included components of the OXPHOS complex essential for the mitochondrial metabolic process (ATP5A, ATP5C1, CYC1, UQCRC1, NDUFS1, and NDUFA9). The OXPHOS complex is another target responsible

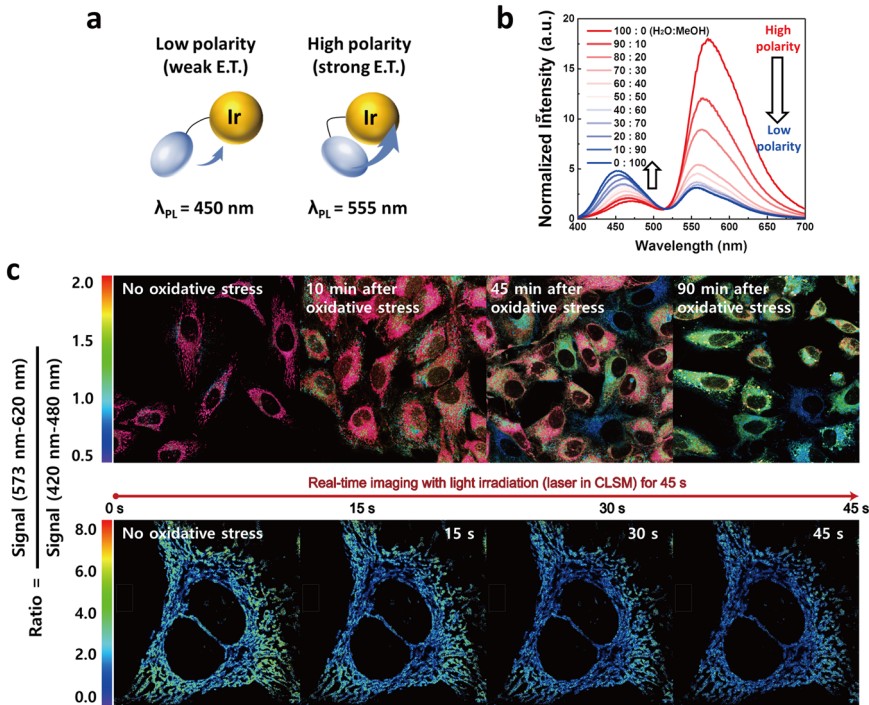

**Fig. 5 Mitochondrial depolarisation by photoactivation of Ir-OA. a** Schematic illustrating polarity dependent energy transfer efficiency changes and (**b**) following ratiometric emission property changes depending on the $H_2O$:MeOH ratio. **c** Ratiometric CLSM imaging of HeLa cells with Ir-OA according to giving oxidative stress. Ratiometric emission was observed 0, 10, 45, and 90 min after oxidative stress exposure (LED array, $\lambda = 400$ nm; light dose = 0.17 J cm$^{-2}$) (top, scale bars = 50 μm), and the mitochondrial polarity change was monitored by the real-time ratiometric imaging during photoactivation within 60 s (bottom, scale bars = 20 μm). The CLSM instrument's laser excited Ir-OA for real-time imaging (bottom). Ratio = (emission of acceptor, $\lambda_{em}$ = 573–620 nm/emission of donor, $\lambda_{em}$ = 420–480 nm). Normal mitochondria with high MMP: red (top) and green (bottom). All imaging was repeated three times independently, and each experiment showed similar results. Source data are provided as a Source Data file.

for possible mitochondrial polarity changes following photoactivation of Ir-OA. The OXPHOS complex is involved in the vast majority of ATP production and H$^+$ efflux/influx between the IMS and the matrix[41,42]. ATP5A1 and ATP5C1 are components of OXPHOS complex V and responsible for H$^+$ influx, which is solely driven by OXPHOS complex V. Therefore, along with Ca$^{2+}$ imbalance, H$^+$ gradient collapse also contributes to mitochondrial depolarisation. Note that the crystal structures of the voltage-dependent anion channel (VDAC1), OXPHOS complex I matrix arm (NDUFS1 and NDUFA9), and OXPHOS complex III (CYC1 and UQCRC1) were described to indicate the O-Met site (Fig. 6d)[43,44]. Moreover, the above-mentioned VDAC1, VDAC3, and OXPHOS complex are closely associated with apoptosis[39,45]. This oxidative stress for each crucial protein could be related to the acceleration of oxidation-induced cell death. Thereby, we conclude that oxidation by Ir-OA photoactivation significantly affects not only mitochondrial depolarisation but also other mitochondrial functions and physiology.

**Monitoring mitochondrial morphology change.** The mitochondrial morphological changes and noticeable swelling are decisive evidence representing cell death progression[46]. We recorded the mitochondrial morphology in real-time by utilising time-lapse imaging during light irradiation (14 mW, 405 nm laser of laser scanning microscopy) (Fig. 7a, b). Interestingly, the shape of most mitochondria became round within 180 s, and the mitochondrial matrix swelling occurred with frequent fission and fusion (Fig. 7a, b and Supplementary Movies 2–4). In our O-Met proteome, several protease and chaperones were oxidised (Fig. 7c), which could damage their functions of eliminating and restoring un-/misfolded mitochondrial proteins. Therefore, it

accumulated damaged proteins and triggered corresponding mitochondrial fission/fusion and swelling. Fission/fusion, known as the protein quality control process, could be related to oxidation of mitochondria/ER proteases and chaperones. Thus, fission/fusion is overloaded due to the accumulation of damaged proteins inside the mitochondria caused by the dysfunction of proteolysis. The phenomenon corresponds to the fact that the proteins involved in fission and fusion were not observed in the O-Met proteome (Fig. 7d). Further, mitochondrial matrix swelling could be explained by the oxidation of OXPHOS and channel proteins leading to an ion imbalance (Fig. 7e). The dysfunction of OXPHOS complex I must increase the NADH level and cause VDAC closure, increasing matrix Ca$^{2+}$ concentration[47,48]; additionally, the Ca$^{2+}$ level becomes elevated by the oxidation of LETM1[46], involved in Ca$^{2+}$ efflux. The accumulation of Ca$^{2+}$ inside the matrix, mitochondrial depolarisation, and ATP depletion caused by dysfunction of OXPHOS may trigger the mitochondrial permeability transition pore (MPTP) opening that accelerates influx of ions, water, and other solutes[46,49,50]. Accordingly, the MPTP opening swells the mitochondrial matrix, leading to cell death by abolishing the OMM to release cell death-inducing factors (Fig. 7e)[46].

## Discussion

Based on phenomenological monitoring and proteomic analysis, we present a promising mechanism from mitochondrial oxidative stress to cell death (Fig. 7f). Primarily, ROS crosslinked proteins around the mitochondria, which increased microviscosity. The viscous environment around the OMM/IMM reduces the diffusion of metabolites and the physiological reaction rate, disrupting metabolite-mediated mitochondrial functions, which include

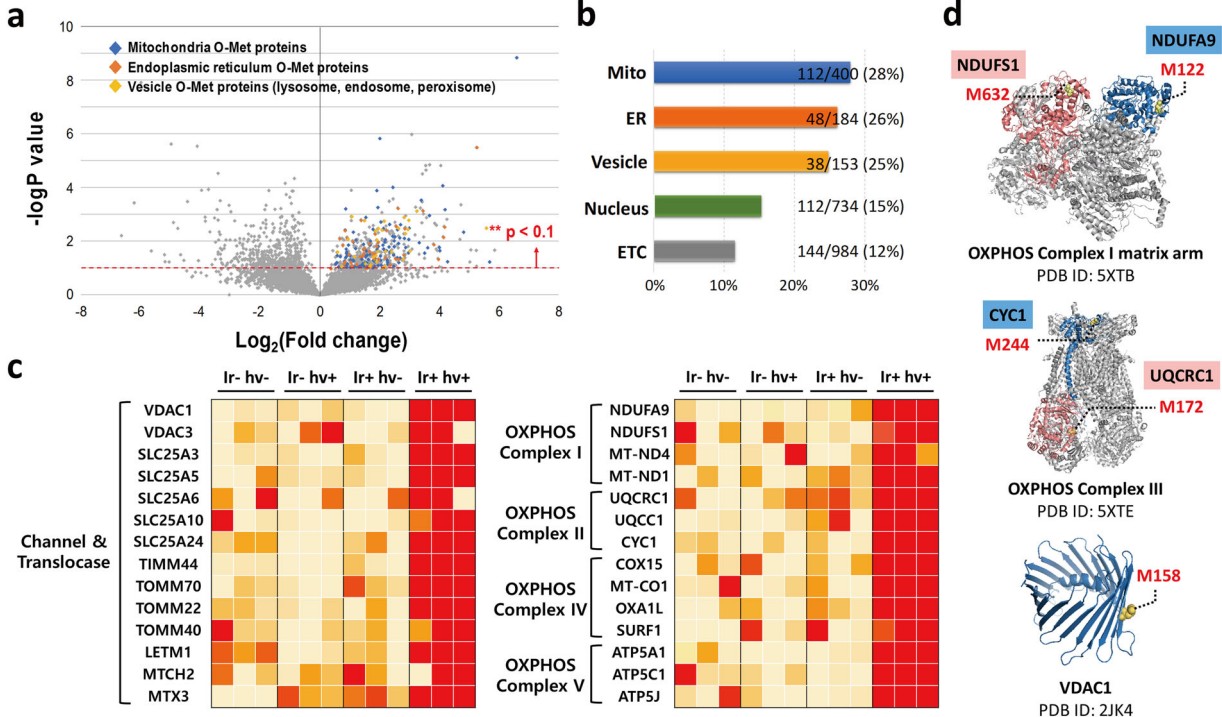

**Fig. 6 Mitochondrial depolarisation related proteomics. a** Quantitatively analysed Volcano plot of oxidised proteome. The substantially oxidised proteome was sorted in the range of log$_2$(Fold change) > 0 and −logP value > 1 (light blue region). **b** Proportion of the oxidised proteome in various organelles. Proteome location was determined using UniProt and mitochondria proteins were cross-checked with Human MitoCarta2.0 dataset consisting of 1158 human genes. The proportion (%) was obtained as ratio between the number of significantly oxidised proteome (−logP > 1) and that of whole oxidised proteome (−logP > 0) in each cell organelle. **c** Heat map with label free quantification (LFQ) values for 28 mitochondrial proteins among 112 substantially oxidised mitochondrial proteins (Supplementary Data). **d** Three representative oxidised proteins; OXPHOS complex I (NDUFS1 and NDUFA9), OXPHOS complex III, and voltage-dependent anion-selective channel 1 (VDAC1). The crystal structures of three proteins from RCSB protein data bank (PBD ID) were visualised and processed with PyMOL.

ATP production, mitochondrial protein synthesis, and ROS reduction. In addition, ROS oxidise proteins (OXPHOS complex and channel & translocase) responsible for maintaining MMP, causing a serious dysfunction in the proteins and resulting in mitochondrial depolarisation. The oxidised proteins induce Ca$^{2+}$ accumulation in the matrix and lead to MPTP opening, triggering noticeable swelling of mitochondria, accompanied by fission and fusion, followed by cell death. Overall, the mitochondrial responses to oxidative stress, such as viscosity enhancement, depolarisation, and subsequent mitochondrial swelling have a synergistic effect on cell death.

Using a molecular design strategy based on intramolecular energy transfer, we report a strong ROS-producing iridium(III) photosensitiser, Ir-OA, inducing mitochondrial oxidation-induced cell death. The photophysical properties of Ir-OA were applied to various imaging techniques to monitor the instantaneous mitochondrial responses to the strong oxidative stress. Interestingly, increased microviscosity and depolarisation were caused by proteins crosslinking around the mitochondria due to oxidative stress and oxidised mitochondrial proteins associated with the channel and translocase, as well as the OXPHOS complex. In addition, we observed noticeable swelling via MPTP opening through Ca$^{2+}$ accumulation, depolarisation, and ATP depletion, which accelerated cell death. Furthermore, oxidation of mitochondria/ER proteases caused accumulation of damaged proteins, leading to mitochondrial fission/fusion. Briefly, we propose a potential mechanism from mitochondrial oxidative stress to cell death using a photosensitiser for phenomenological observations and proteomic analyses. These results suggest a way in which mitochondrial photosensitisation affects cellular

survival. We hope that these results will contribute to providing a fundamental understanding of mitochondrial oxidation-related diseases, as well as cancer therapeutics inducing oxidative stress.

## Methods

**General**. The details of synthesis, characterisation, photophysical properties, time correlated single photon counting (TCSPC), cell imaging, cell viability test, and western blot are provided in the Supplementary Information.

**LC-MS/MS analysis for methionine oxidised proteome**. HEK293T cells were grown in 6-well plates in DMEM supplemented with 10% FBS, 50 units/mL penicillin, and 50 μg/mL streptomycin under 37 °C and 5% CO$_2$ conditions. Grown cells were incubated with 5 μM Ir-OA for 1 h, and the samples were irradiated with LED array (400 nm, 3.75 mW cm$^{-2}$ for 1 min, 225 mJ cm$^{-2}$). Note that four different samples with negative controls were prepared (#1: Ir-/hv-, #2: Ir-/hv +, #3: Ir + /hv-, #4: Ir + /hv+). Irradiated cells were lysed by RIPA buffer for 20 min at 4 °C. After centrifugation (16,000 × $g$, 10 min, 4 °C), the supernatant protein lysis solution was denatured and separated by SDS-PAGE gel electrophoresis. Protein loading quantity was quantified using the BCA assay, and 50 μg of protein was loaded for each of the samples. For whole protein analysis with 'Shot-Gun' method, gel electrophoresis was proceeded up to approximately 1 cm length of the lane. The gel was stained with Coomassie blue for 2 h and washed for 6–12 h with destaining solution (H$_2$O, methanol, and acetic acid in a ratio of 50/40/10% (v/v/v)). Each clearly stained lane was divided into six parts according to protein size; these were further chopped up to small cubic shapes (approximately 1 mm × 1 mm × 1 mm) for efficient in-gel digestion. Each of the six parts was transferred to 1.5 mL low binding microtubes (Eppendorf, Hamburg, Germany). Gels were washed with 150 μL triple distilled water for 5 min on the shaker (in triplicate). Next, 150 μL of 0.1 M ammonium bicarbonate (ABC) was added to the tubes and washed on the shaker for 5 min (in triplicate). The tubes were incubated with 1:1 mixture of 0.1 M ABC and acetonitrile on the shaker for 5 min (in triplicate). The above step was repeated with 100% acetonitrile. (in triplicate). The whole washing process with ABC solution to acetonitrile gradient was repeated once again. Residual solvent after removal of the final acetonitrile was completely dried with a speed-vac. After the drying process, a 150 μL solution mixture of 10 mM dithiothreitol (DTT) and

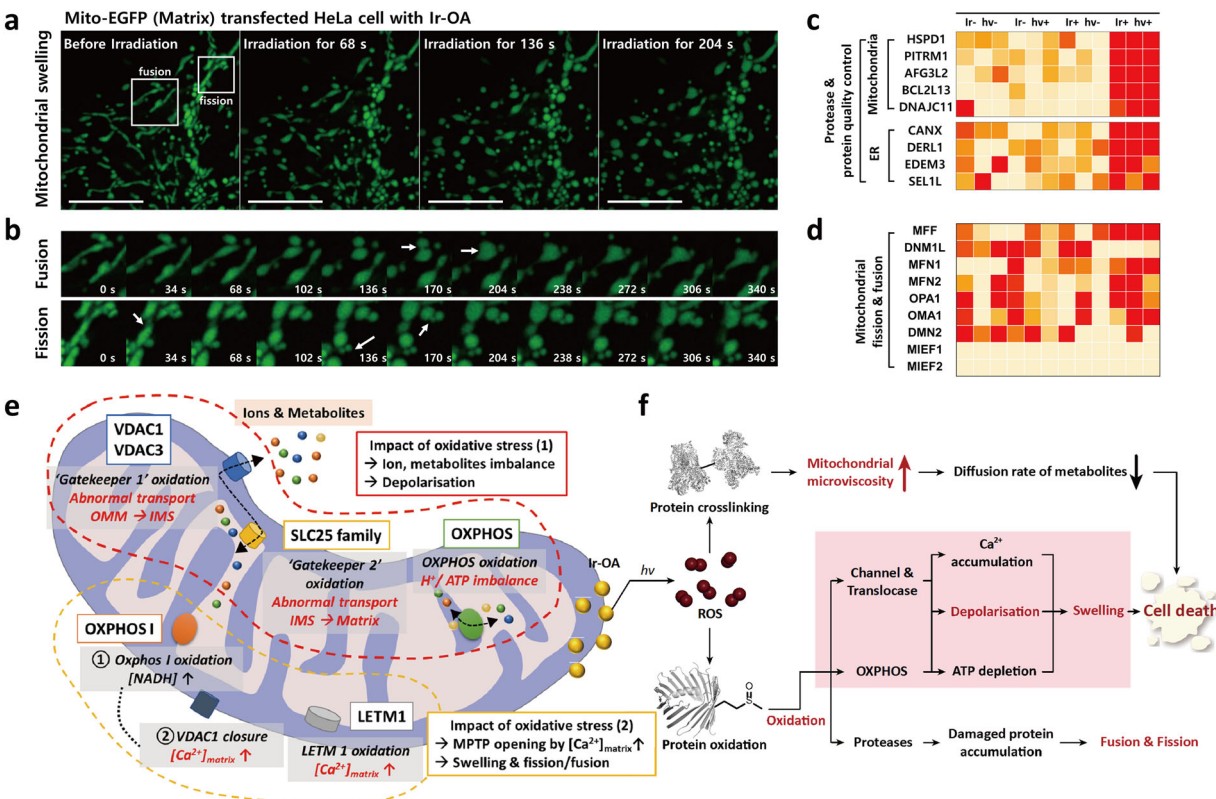

**Fig. 7 Mitochondria morphology monitoring and proposed cell death mechanism. a** Time-lapse Airyscan 2 images of HeLa cells with Ir-OA before light irradiation (14 mW, 405 nm laser of laser scanning microscopy) (left) and after irradiation for 204 s (right). To monitor morphological changes, the mitochondrial matrix was transfected by Mito-EGFP (green signal) (scale bars = 10 μm). White boxes indicate mitochondrial fission and fusion. The experiment was repeated three times independently, and each experiment showed similar results. **b** Enlarged time lapse images (0–340 s) of white boxes from Fig. 7a. Along with fission and fusion, mitochondria swelling was also observed. **c** Investigation of proteases and mitochondrial fission and fusion-related protein oxidation. Heat map diagram for O-Met proteins related to mitochondrial protein quality control, in addition to (**d**) fission, and fusion. **e** Brief description of a mechanism for mitochondrial environment change. This illustrates the impact of mitochondrial oxidative stress based on proteomic analysis of mitochondrial oxidative stress. **f** Mechanistic description of mitochondrial oxidation-induced cell death based on phenomenological observations and proteome analyses.

100 mM ABC for reduction was added into each microtube, and the mixture was incubated using the ThermoMixer (79 × *g*, 60 min, 56 °C) (Eppendorf, Hamburg, Germany). The above solution for reduction was replaced with a 150 μL solution mixture of 55 mM iodoacetamide (IDA) and 0.1 M ABC for protein alkylation and the tubes were shaken on ThermoMmixer (79 × *g*, 30 min, 25 °C) in the dark. After alkylation was completed, 100 mM ABC was added, and the tube was shaken for 5 min. Next, 0.1 M ABC was replaced with a 1:1 mixture of 0.1 M ABC and acetonitrile. The tubes were also shaken for 5 min and the mixture solution was substituted with only acetonitrile. ABC washing steps after alkylation were repeated once again. The last residual acetonitrile was removed using a micro-pipette and the tubes were strictly dried again by speed-vac. Then, 25 ng/μL Trypsin Gold mass spectrometry grade (V5280; Promega, WI, USA) in 0.05 M ABC was added to microtubes and incubated on the ThermoMixer (20 xg, 12–18 h, 37 °C). The microtubes were vortexed and supernatant was transferred to other new microtubes. The remaining gels in old tubes were washed with a 2:1 mixture of 5% formic acid and acetonitrile and the supernatant was transferred again to the same tubes. The supernatants were fully dried by using a speed-vac. Prepared peptide samples were analysed by Q Exactive Plus orbitrap mass spectrometry (Thermo Fisher Scientific, MA, USA) equipped with a nanoelectrospray ion source. To separate the peptide mixture, we used a C18 reverse-phase HPLC column (500 mm × 75 μm ID) with an acetonitrile/0.1% formic acid gradient from 2.4–28% for 150 min at a flow rate of 300 nL/min. For MS/MS analysis, the precursor ion scan MS spectra (*m/z* 400–2000) were acquired in the Orbitrap at a resolution of 70,000 at *m/z* 400 with an internal lock mass. The 20 most intensive ions were isolated and fragmented by high-energy collision induced dissociation (HCD). The experiment was performed in triplicate. LC-MS/MS data acquisition software Xcalibur (ver. 4.1.31.9) from Thermofisher Scientific was utilised.

**LC-MS/MS data processing**. The Sequest Sorcerer platform (Sagen-N Research, San Jose, CA) was utilised to analyse the prepared LC-MS/MS samples and found the *Homo sapiens* protein sequence database (42284 entries, UniProt (http://www.uniprot.org/)) with a fragment ion mass tolerance of 1.00 Da and a parent ion

tolerance of 10.0 ppm. The Sequest specified carbamidomethylation of cysteine as a fixed modification and the oxidation of methionine and acetyl of the *N*-terminus as variable modification. The MS/MS-based protein and peptide identifications were validated by Scaffold (Version 4.9.0, Proteome Software Inc., Portland, OR). The peptides were identified if they could be established at greater than 92.0% probability to reach a false discovery rate (FDR) less than 1.0% by the Scaffold Local FDR algorithm. In addition, proteins were identified if they could be established at greater than 88.0% probability to reach an FDR less than 1.0% and contained at least two identified peptides; protein probabilities were assigned by the Protein Prophet algorithm[51]. The proteins hardly distinguished by MS/MS analysis were grouped to satisfy the principle of parsimony. Proteins were annotated with GO terms of NCBI (downloaded April 14, 2019)[52]. Based on proteome data, the top 3 precursor intensity value was utilised for label-free quantification (LFQ) of methionine oxidation. LFQ intensity values were log2-transformed and empty values for each condition (not reproducibly detected oxidised-proteome) were filled by imputed values representing a normal distribution around the detection limit. Initially, we obtained the intensity distribution of mean and standard deviation. Based on total matrix, a new distribution was created by Gaussian distribution with a downshift of 1.8 and width of 0.3 standard deviations. All processes were progressed using the Perseus software platform from Max Planck Institute of Biochemistry.

**Reporting summary**. Further information on research design is available in the Nature Research Reporting Summary linked to this article.

## Data availability

The authors declare that the data supporting the findings of this study are available within the article and its Supplementary Information. The protein information utilised in this study is available from the RCSB protein data bank (PBD ID) (http://www.rcsb.org/pdb/) and the *Homo sapiens* protein sequence database (42284 entries, UniProt (http://www.uniprot.org/)). The mass spectrometry proteomics data that support the findings of this study have been deposited to the ProteomeXchange Consortium via the PRIDE

partner repository with the dataset identifier PXD022163. Extra data are available from the corresponding author upon reasonable request. Source data are provided with this paper.

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

## Acknowledgements

The authors gratefully acknowledge the financial support provided by Ulsan National Institute of Science and Technology (UNIST) (Research Fund 1.200030.01), the National Research Foundation of Korea (NRF) (grant 2016R1A2B4009239, and 2017M3A7B4052802), the Technology Development Program to Solve Climate Changes

of the NRF funded by the Ministry of Science, ICT and Future Planning (grant 2016M1A2A2940910, and 2017M1A2A2087813), and New Renewable Energy Core Technology Development Project of the Korea Institute of Energy Technology Evaluation and Planning(KETEP) granted financial resource from the Ministry of Trade, Industry and Energy, Republic of Korea (grant No. 20183010013900). J.S.N. is thankful to the support of the ASAN Foundation Biomedical Science scholarship. C.G.L. acknowledges the support from the Global Ph.D. fellowship program through the National Research Foundation of Korea (NRF) funded by the Ministry of Education (grant NRF-2018H1A2A1061237).

## Author contributions

C.L., J.S.N. and T.-H.K. wrote the manuscript. All authors have given approval to the final version of the manuscript. C.L. and J.S.N. contributed equally to this work. C.L. synthesised and characterised all compounds. C.L. and J.S.N. conceived cell imaging, western blot and mass analysis. C.L. obtained and processed images and J.S.N. analysed western blot and oxidised proteome data from mass spectrometry. All mass analysis was reviewed and supported by J.K.S. and H.-W.R. C.G.L. carried out the MTT-assay. M.P. conducted the TCSPC analysis. C.Y. prepared all plasmids and supported proteomic analysis. All authors discussed the results and commented on the manuscript. T.-H.K. supervised the project.

## Competing interests

The authors declare no competing interests.
