## [Peer Review File · Nature Communications]

Reviewers' comments:

Reviewer #1 (Remarks to the Author):

There are a lot of interesting data in this paper but I was not convinced by several of the claims. In the conclusions are two of them.

They claim "a comprehensive mechanism from mitochondrial oxidative stress to cell death based on phenomenological observations and proteomic analyses by employing Ir-OA." The mechanism here is likely to be specific to the probe used since it will preassociate with biomolecules in advance of singlet oxygen generation.

Also they claim that "this research will serve to overcome the hurdles currently restricting advancement in human mitochondria-related diseases." is also not convincing. How will it do that? The identification of oxidative damage to proteins is somewhat limited- to oxidation of methionine and cross linking. Literature reports also describe other sites of singlet oxygen induced damage and the nature of the cross-linking is not well described (are they assuming it all involves tyrosines (Fig 6)

Specific comments.

L48

"mitochondrial viscosity, polarity, morphology, pH, and temperature)"

Not easy to understand the definition of 'mitochondrial viscosity' since mitochondria are heterogeneous

Similarly for the other parameters.

L73

"caused an ROS deficiency"

Meaning?

L88

"Ru(bpy)₃"

[Ru(bpy)₃]²⁺ ?

L98

"Accordingly, Ir-OA can be employed as a remarkable photosensitizer to cause strong oxidative stress to live cells"

cannot be said at this point – no cell data presented yet.

P103

"which is fluorescent"

which is a fluorescent

L114

"the major population of Ir-OA is not in"

the major population of Ir-OA is not in

At least the population that is luminescent (others may be quenched)

L119

"causing protein dysfunction, microenvironment changes, and cell death"

Perhaps

"we conducted the MTT assay for quantitative cytotoxicity"

The MTT assay is a poor choice of cytotoxicity assay for cells with damaged mitochondria since it

relies on mitochondrial succinate dehydrogenase and is dependent on mitochondrial respiration.

L134

"mitochondrial viscosity increment"

Meaning?

L153

"implying that crosslinked BSA triggered enhanced viscosity"

What is the evidence for BSA cross-linking under these conditions? And what is the extent?

L160

"this crosslinked eGFP signal"

Presumably refers to a band on a gel?

L165

"generated in the ER ($\eta = 60.3\%$),"

So the major effects of photoactivation are in the cytoplasm (in the ER) and not in mitochondria-correct?

L167

"protein aggregates surrounding the mitochondria"

So is this due to cross-linked proteins in the cytoplasm ?

How elevated are the levels of the proteins after transfection compared to their normal levels?

Does transfection change the potency of the photosensitiser?

L202

"Protein oxidation by photoactivated Ir-202

OA occurred in the mitochondria, ER, and peroxisome"

Not in the cytoplasm – not even in those proteins surrounding the mitochondria ?

(as suggested earlier)

L227

"were provided with an indication of the O-Met site ("

Meaning?

L229

"proteases oxidation occurs"

protease oxidation occurs

L230

"is not properly restored"

After what? cell division?

L230

"This oxidative stress for each crucial protein accelerates oxidation-induced cell death"

Perhaps

"light irradiation"

Details of irradiation should be mentioned in the text

L257

"ROS oxidises proteins"

ROS oxidises proteins

L266

"we reported"

we report

P288

How long were the cells irradiated for?

P332

"acetyl of the n-terminus"

acetyl of the N-terminus

L393

Ref 22 not properly formatted

L490

"Localisation matter of Ir-OA in living cells"

Rephrase

L494

Figure 2C seems to show Ir-OA spanning a width of 1.5 μm (1,500 nm). This is much bigger than a mitochondrion (ca. 300 nm) so it is not clear what is being observed.

What is missing are dark and light IC50 values and an indication of how many cells are damaged under the conditions of the confocal imaging.

L115: "This super-resolution imaging result supports that the sub-mitochondrial localisation of Ir-OA might be from IMM spreading to the IMS and outer mitochondrial membrane (OMM) (Figure 2C)."

In several places protein cross-linking is shown as coupled tyrosines – what is the evidence for that in the present work?

SI

L155

"Pd(II)OAc"

Pd(II)(OAc)₂?

Ir-OA: CHN analysis? HPLC purity?

Cell culture

Needs to be stated how the compound was added to the cells. Dissolved in DMSO?

The switch between HeLa, A549 and HEK cells needs to be explained. I did not see A549 mentioned at all in the text (just SI). It cannot be assumed that different cell types will respond in the same way.

Basic IC50 data on them all are missing.

Are any of the Ns protonated at pH 7 in the side arm of Ir-OA?

Reviewer #2 (Remarks to the Author):

This manuscript reports an iridium complex that can produce ROS upon irradiation, and at the same time report on viscosity changes in the immediate environment. Importantly, a negative control analogue is used throughout the paper. The authors have used this complex in detailed

biological studies of cell death caused by mitochondrial oxidative stress. This work encompasses a range of elegant studies across chemistry and biology, and is therefore appropriate for publication in this journal, after the following amendments:

1. While the negative control does provide some information, the authors also need to include irradiation alone as another negative control in their studies. It has been reported that 400 nm light does induce ROS, so studies should be carried out to ascertain whether this contributes significantly to the results reported here. This is particularly important for the use of DCFDA: this is now largely rejected as a meaningful reporter of ROS, and can be photoactivated itself, so the control experiment is required here.
2. Both IrOA and IrOC are referred to as photosensitisers throughout, but IrOC in fact cannot act as a photosensitiser – that is why it is the negative control. The terminology should therefore be amended.
3. The live-dead assay described here does not provide such valuable information as no statistical data is provided, and the red and green signals are not correlated. This assay should be repeated by flow cytometry to provide these data. Furthermore, an apoptosis-necrosis assay could also be done at the same time to confirm the hypothesis that photoactivation of IrOA induces apoptosis.
4. For both the FLIM studies in cells, and the BSA studies in cuvette, statistical information must be provided, otherwise it is impossible to ascertain whether these results are significant. This is particularly important for the cells, where a change from 844 to 911 ns could easily not be significant.
5. There is clearly some complex that is not in the mitochondria, and the authors conclude some effect of the complex on the ER and peroxisomes, among all organelles, but there is no evidence for this. Since the GFP-labelled organelles are available, co-localisation studies should be carried out of IrOA with all 4 of these labelled cells.
6. Figure S18 – the figure caption must specify what the merged images correspond to.

Reviewer #3 (Remarks to the Author):

This paper describes strategy based to monitor intramolecular energy transfer to produce mitochondrial oxidation-induced cell death. The synthesis of the reagent and all supporting analyses is well-documented in the supporting information.

Various imaging techniques were then employed to monitor the mitochondrial responses to this strong oxidative stress. Microviscosity and depolarisation were caused by protein crosslinking around the mitochondria due to the application of this reagent. Oxidised methionine residues were used to report upon the mitochondrial proteins that were affected by the process and a label free proteomics approach was adopted to tabulate these results. Many of the observations were interpreted in terms of effects associated with the channel and translocase proteins as well as the OXPHOS chain.

The experiments are on the whole well-designed and well-executed and I recommend publication following attention to some minor issues:

Points to address:

1. Figure 5 e and a number of the other figures are hard to follow – primarily since the font used for labelling is so small and the details are not well described in the legends.
2. It is interesting that the proteins for which effects were observed largely correlated with those that are present at the highest abundance.
3. Figure S20 to me made the paper more comprehensible – I would suggest promoting this to the introduction or conclusion of the paper.
4. I think the concluding remarks oversell the paper too strongly – this a good well-executed study that I recommend for publication but am not sure that it will go far in solving problems in mitochondrial diseases – this link was not apparent at least to me.

Reviewers' comments:

Reviewer #1 (Remarks to the Author):

There are a lot of interesting data in this paper but I was not convinced by several of the claims. In the conclusions are two of them.

Q1. They claim “a comprehensive mechanism from mitochondrial oxidative stress to cell death based on phenomenological observations and proteomic analyses by employing Ir-OA.”

The mechanism here is likely to be specific to the probe used since it will preassociate with biomolecules in advance of singlet oxygen generation.

Response:

We agree on your comment. The suggested mechanism focused on the photoactivation of photosensitiser as ROS generator. So, we revised the sentence like below. Thank you for the sharp comment.

Page 2 Line 30

“Consequently, we suggest a comprehensive mechanism ~”

→ “Consequently, we suggest a potential mechanism ~”

Page 14 Line 333

“we propose a comprehensive mechanism from mitochondrial oxidative stress to cell death based on phenomenological observations and proteomic analyses by employing **Ir-OA**.”

→ “we propose a potential mechanism from mitochondrial oxidative stress to cell death using a photosensitiser for phenomenological observations and proteomic analyses.”

Q2. Also they claim that “this research will serve to overcome the hurdles currently restricting advancement in human mitochondria-related diseases.” is also not convincing. How will it do that?

Response:

→ We really appreciate this valuable comment. To make our conclusion reliable, we revised the related sentence and paragraph in the introduction and conclusion sections like below.

Page 2 Line 19

“Mitochondrial oxidation-induced cell death, a physiological process implicated in ageing and the pathogenesis of various diseases, has been ~”

→ “Mitochondrial oxidation-induced cell death, a physiological process triggered by various cancer therapeutics to induce oxidative stress on tumours, has been ~”

Page 3 Line 40

“~ is essential to understanding cellular ageing and the pathogenesis of various diseases.”

→ “~ is essential to understanding and improving cancer therapeutics based on oxidative damage to tumours.”

Page 4 Line 66

“ ~ will pave the way to understanding cell ageing, death, and pathogenesis triggered by oxidative stress.”

→ “~ will aid the understanding of cancer therapeutics induced mitochondrial oxidative stress.”

Page 15 Line 335

“These results elucidate the pathology of mitochondrial oxidation-related diseases and provide a fundamental understanding of mitochondrial oxidation-induced ageing and death. We believe that this research will serve to overcome the hurdles currently restricting advancement in human mitochondria-related diseases.”

→ “These results suggest the way in which mitochondrial photosensitisation affects cellular survival. We hope that these results will aid in providing a fundamental understanding of mitochondrial oxidation-related diseases as well as cancer therapeutics inducing oxidative stress.”

Q3. The identification of oxidative damage to proteins is somewhat limited- to oxidation of methionine and cross linking. Literature reports also describe other sites of singlet oxygen induced damage and the nature of the cross-linking is not well described (are they assuming it all involves tyrosines (Fig 6)

Response:

We are sincerely sorry for the misleading figures (#4 and 6). It was reported by Kodadek group that ruthenium photosensitiser triggers crosslinked proteins via the formation of dityrosine (Fancy and Kodadek. *Proc. Natl. Acad. Sci. USA* **96** (11), 6020-6024 (1999)). In addition, we also found that the iridium photosensitiser induced dityrosine bond as well (Nam et al. *J. Am. Chem. Soc.* **138**, 10968-10977 (2016)). Thus, we assumed that the crosslinked proteins formed by photosensitisation of **Ir-OA** would be induced via dityrosine crosslinking. However, as you mentioned, it was not adequately supported by experiment results. Therefore, we experimented for identifying dityrosine bond formation inside cells by using a dityrosine antibody as below. According to the immunostaining imaging results, we could observe the enhancement of dityrosine bond signal in the presence of **Ir-OA** and light irradiation (1.25 J cm⁻², 400 nm blue LED). However, the enhancement was not significant, so we concluded that the dityrosine bond was involved in the photo-crosslinking, but it does not seem the dominant factor of the photo-crosslinking. Thus, we removed related figures (Figure 4A and 6C). In our further research, we are on the way to find out specific chemical bonds involved in the photosensitization-induced crosslinking mechanism.

Figure R1. (Left) Images of dityrosine antibody–anti-Mouse Alexa Fluor® 647 with condition of +**Ir-OA**/+hv, +**Ir-OA**/-hv, and -**Ir-OA**/+hv. HeLa cells with each condition are irradiated by LED array ($\lambda = 400$ nm, 1.25 J cm⁻²), then were fixed and immunostained by anti-dityrosine and Alexa 647 dye. (Right) Averaged signal intensity of anti-dityrosine of 20 HeLa cells.

Next, methionine has been known as easily-oxidised amino acid under oxidative stress (Lee et al. *Biochimica et Biophysica Acta*. **1790** (11), 1471-1477 (2009)). Besides, the oxidised methionine shows dramatic polarity difference compare to the other oxidised amino acids (Tyr, His, Cys) as described below, which could disturb protein functions (Stadtman et al. *Biochimica et Biophysica Acta*. **1703** (2), 135–140 (2005)). Thus, we supposed that the methionine-oxidised proteins represent a loss of their original function. So, we chose methionine as a target amino acid to identify dysfunctional proteins.

Figure R2. Density functional theory (DFT) calculation was performed with B3LYP functional and the basis set of 6-311G(d,p). This figure shows neutral and oxidised amino acids with electrostatic surface potential (ESP), showing redder electron cloud with positive potential and bluer electron cloud with negative potential. The sulfur atom of methionine (Met) gets much positive potential as it oxidised to have thionyl (S=O) group, which implies that methionine oxidation can induce protein misfolding and dysfunction.

Specific comments.

We really express the gratitude for all specific comments including typo and logical issues. Thus, we commissioned a proofreading service for additional English correction. All your comments were constructive to improve this manuscript. The responses to each comment are described below.

Q3: L48 “mitochondrial viscosity, polarity, morphology, pH, and temperature)”

Not easy to understand the definition of ‘mitochondrial viscosity’ since mitochondria are heterogeneous. Similarly for the other parameters.

Response:

As you mentioned, mitochondria are heterogeneous. However, most of them show uniform changes in their characteristic such as polarity, morphology, pH, viscosity, and temperature when external stress (such as oxidative stress) occurs. Thus, ‘the biological phenomena’ means the simultaneous change in characteristics and environments of most mitochondria in response to oxidative stress.

Specifically, it is also difficult to define mitochondrial viscosity because of its complex structures. The viscosity in this manuscript means the local viscosity around mitochondrial membrane, because **Ir-OA** is located on the mitochondrial membrane. Thus, we revised the part of the manuscript as below:

Page 3 Line 50

“~ due to the absence of a chemical tool to analyse the biological phenomena (*i.e.* mitochondrial viscosity, polarity, morphology, pH, and temperature) that occur in mitochondria in response to oxidative stress.”

→ “~ due to the absence of a chemical tool to analyse the biological phenomena (*i.e.*, environmental changes in mitochondrial surroundings in terms of viscosity, polarity, morphology, pH, and temperature) that occur in mitochondria in response to oxidative stress.”

Q4: L73

“caused an ROS deficiency”

Meaning?

Response:

Page 4 Line 77

“~, which caused an ROS deficiency.”

→ “resulting in a limited ROS production.”

Q5: L88

“Ru(bpy)3”

[Ru(bpy)3]2+ ?

Response:

Page 5 Line 93

“Ru(bpy)3”

→ “[Ru(bpy)3]²⁺”

Q6: L98

“Accordingly, Ir-OA can be employed as a remarkable photosensitiser to cause strong oxidative stress to live cells”

cannot be said at this point – no cell data presented yet.

Response:

This sentence was followed by the H₂DCF-DA assay in page 5 line 97, representing intracellular ROS generation inside cells (Supplementary Figure 11). Because the result of H₂DCF-DA assay represents that the **Ir-OA** produce much more ROS inside cells, then we mentioned that **Ir-OA** could be remarkable photosensitiser inducing strong oxidative stress rather than **Ir-OC**. For the clarity, we revised the manuscript as below.

Page 5 Line 102

“Accordingly, **Ir-OA** can be employed as a remarkable photosensitiser to cause strong oxidative stress to live cells ~”

→ “According to the intracellular ROS assay (Supplementary Figure 11), **Ir-OA** can be employed as an effective photosensitiser to induce strong oxidative stress to live cells”

Q7: P103

“which is fluorescent”

which is a fluorescent

Response:

Page 5 Line 111

“which is fluorescent”

→ “which is a fluorescent”

Q8: L114

“the major population of Ir-OA is not in”

the major population of Ir-OA is not in

At least the population that is luminescent (others may be quenched)

Response:

Page 6 Line 121

“~ the major population of **Ir-OA** are not in ~”

→ “~ the major population of luminescent **Ir-OA** is not in ~”

Q9: L119

“causing protein dysfunction, microenvironment changes, and cell death”

Perhaps

Response:

Page 6 Line 127

“~, causing ~”

→ “~, which may cause ~”

Q10: “we conducted the MTT assay for quantitative cytotoxicity”

The MTT assay is a poor choice of cytotoxicity assay for cells with damaged mitochondria since it relies on mitochondrial succinate dehydrogenase and is dependent on mitochondrial respiration.

Response:

We appreciate this constructive comment. We experimented a cell viability test by using CCK-8 assay because this assay utilizing WST-8 is independent of mitochondrial metabolism (Figure 3A). As described below, the **Ir-OA** shows phototoxicity over 8 μM concentration, while **Ir-OC** is not cytotoxic under weak light exposure ($<0.5 \text{ J cm}^{-2}$). Notably, the phototoxicity of **Ir-OA** derived by CCK-8 assay is lower than that derived by MTT assay. This result implies that CCK-8 assay is appropriate for estimate phototoxicity of photosensitiser targeting mitochondria. Thus, we changed the cell viability data from the

MTT assay to CCK-8 assay of Figure 3, and added supplementary table containing IC₅₀ of each assay and corresponding sentences as below. Thank you for your incisive comment.

Page 6 Line 135

→ “Notably, the IC₅₀ value of **Ir-OA** obtained using the CCK-8 assay was considerably higher than that obtained using the MTT assay (Supplementary Table 2). This is because the formed formazan in the MTT assay depends on the mitochondrial dehydrogenase, while CCK-8 is activated by dehydrogenases from whole cells. Thus, the CCK-8 assay is more appropriate and reliable in measuring the phototoxicity of photosensitisers targeting the mitochondria” (added)

Figure 3. Identification of phototoxicity effect from iridium(III) complexes. A) CCK-8 assay for quantifying the cytotoxicity of **Ir-OA** and **Ir-OC** with or without light irradiation for HeLa cells. All error bars = s.d. (n=3). Conditions: photosensitiser iridium(III) complexes incubation time = 2 h, light source = 400 nm light LED array, light dose: 0.08 J cm⁻², 0.17 J cm⁻², and 0.25 J cm⁻². B) Live/dead assay for verifying phototoxicity of **Ir-OA** and **Ir-OC**. At 6 and 20 hours after light irradiation, dead and live cells were stained using propidium iodide

(PI, red) and Calcein AM (green), respectively (scale bars = 200 μm). Conditions: [Iridium(III) complex] = 8 μM , light source = 400 nm light LED array, light dose = 0.255 J cm^{-2} . C) Representative Calcein AM vs PI flow cytometry plot for HeLa cells incubated with iridium complexes. D) Representative Annexin V vs PI flow cytometry plot for HeLa cells incubated with iridium complexes. Q1 (right top): late apoptotic/necrotic cells, Q2 (right bottom): early apoptotic cells, Q3 (left bottom): viable cells, and Q4 (left top): necrotic cells. Conditions for flow cytometry: [Ir complex] = 8 μM , light source = 400 nm light LED array, light dose = 0.25 J cm^{-2} . Source data are provided as a Source Data file.

Supplementary Table 2. Quantitative phototoxicity of **Ir-OA** according to irradiation energy.

Irradiation energy	IC_{50}		
	0.085 J/ cm^2	0.170 J/ cm^2	0.255 J/ cm^2
MTT assay (n=4)	2.25 μM	1.92 μM	1.60 μM
CCK-8 assay (n=4)	15.5 μM	10.2 μM	7.40 μM

The IC_{50} values are measured by mitochondrial function-dependent (MTT assay) and independent (CCK-8) assay. The assays were triplicate at various irradiation energy (0.085, 0.170, and 0.255 J cm^{-2})

Q11: L134

“mitochondrial viscosity increment”

Meaning?

Response:

The sentence mentions that oxidative stress triggers an increase of cytoplasmic and mitochondrial viscosity based on reference 16 and 31. The reference 16 used Iridium(III) complex for detecting mitochondrial viscosity in response to oxidative stress. However, they did not identify the specific site (e.g. mitochondrial matrix, IMS, IMM, or OMM) where viscosity increase, so we mentioned ‘mitochondrial viscosity’. The reference 31 showed that the cytoplasmic viscosity increased by photoactivation of a photosensitiser (porphyrin dyes). Thus, we revised the manuscript as below.

“ROS and mitochondrial oxidative stress triggers microenvironment changes—mitochondrial viscosity increment^{16, 31} and mitochondrial depolarisation^{15, 32,}”

→ “ROS and mitochondrial oxidative stress triggers microenvironment changes, which include increased cytoplasmic and mitochondrial viscosity^{16, 31} and mitochondrial depolarisation^{15, 32,}”

Q12: L153

“implying that crosslinked BSA triggered enhanced viscosity”

What is the evidence for BSA cross-linking under these conditions? And what is the extent?

Response:

We proceeded BSA crosslinking experiments with **Ir-OA** and anti-BSA as described in Supplementary Information. The Western blot result showed that crosslinked BSA signals (o: oligomer, t: trimer, d: dimer) were enhanced by photosensitisation of **Ir-OA**. However, the enhancement was not very significant. So, we reconstructed the BSA experiment; we measured lifetime of **Ir-OA** depending on the BSA concentration to estimate how much the lifetime of **Ir-OA** varies with the local concentration of protein around mitochondria (Supplementary Figure S17). Detailed response is in the Q4 of reviewer 2.

	#1	#2	#3	#4	#5	#6	#7
20 μM Ir-OA	+	-	+	+	+	+	+
1 mM APS	-	-	+	+	-	-	-
light	-	5 min	1 min	5 min	5 min	10 min	20 min
BSA	+	+	+	+	+	+	+

Figure R3. In vitro bovine serum albumin (BSA) photo-crosslinking analysis. BSA has self-oligomerization property in accordance that oligomerized peak is shown in the negative control. The signal of multimerized bands (d, t, and o) becomes intense after photo-irradiation in 5- and 10-min. Note that in vitro photo-crosslinking is less efficient than in cell, which indicates the photo-crosslinking will be accelerated by surrounding biomolecules. Condition: [**Ir-OA**] = 20 μ M, BSA loading quantity = 5 μ g. o = oligomer, t = trimer, d = dimer, m = monomer.

Q13: L160

“this crosslinked eGFP signal”

Presumably refers to a band on a gel?

Response:

You're right. For clear understanding, we revised the sentence like below:

Page 9 Line 190

“this crosslinked eGFP signal was not observed ~”

→ “the signal on a gel with anti-EGFP did not change after photoactivation of **Ir-OA** ~”

Q14: L165

“generated in the ER ($\eta = 60.3\%$),“

So the major effects of photoactivation are in the cytoplasm (in the ER) and not in mitochondria- correct?

Response:

The major effects of photoactivation were observed on the mitochondrial membrane (polarity and morphology) rather than cytoplasm and ER, although the crosslinking efficiency of Sec61b-EGFP is higher than that of Mito-EGFP as you pointed out. The reason for the relatively low crosslinking efficiency of Mito-EGFP can be explained by detailed localisation of the **Ir-OA** as below (Figure R4).

Most **Ir-OA** is probably located on the outer surface of the mitochondrial membrane (IMM and OMM). Thus, the Sec61b-EGFP on ER membrane could contact to **Ir-OA** directly, while the contact between Mito-EGFP in the mitochondrial matrix and the **Ir-OA** is limited by

IMM. Therefore, the crosslinking efficiency of Mito-EGFP can be lower than that of Sec61b-EGFP. All specific localisation of each EGFPs is described by confocal microscopy imaging (revised Figure 4 and Supplementary Figure 19)

Figure R4. Predicted distribution of the **Ir-OA**, Mito-EGFP, and Sec61b-EGFP. The **Ir-OA** is located on the outer face of IMM and OMM, thus they can directly contact to the Sec61b-EGFP, but they cannot efficiently contact to the Mito-EGFP, which cause difference of crosslinking efficiency between Mito-EGFP and Sec61b-EGFP.

So, we added our manuscript the sentence below:

Page 9 Line 203

“Notably, the crosslinking efficiency of ER membrane proteins was more significant than that of mitochondrial matrix proteins and peroxisome proteins because **Ir-OA** is located on the outer surface of the OMM and IMM. The ER membrane proteins were in direct contact with **Ir-OA**, while the contact between the proteins in the mitochondrial matrix and **Ir-OA** is limited by the IMM. Therefore, the **Ir-OA** molecules triggered relatively more protein crosslinking in the ER membrane than that in the mitochondrial matrix.”

Q15: L167

“protein aggregates surrounding the mitochondria”

So is this due to cross-linked proteins in the cytoplasm?

Response:

The formation of ‘protein aggregates’ include crosslinked and damaged (oxidised) proteins of the mitochondrial matrix, membrane, IMS, ER membrane, peroxisome, and maybe cytoplasm. As shown in Figure 4, the proteins surrounding mitochondria (i.e., proteins of Mitochondrial matrix, ER membrane, and peroxisome) are mainly crosslinked by photoactivation of the **Ir-OA**. All these aggregated proteins can induce a viscous environment surrounding mitochondria. So, we revised the sentence like below:

Page 9 Line 210

“The protein crosslinking could be a possible way to increase viscosity in cells by inducing protein aggregates surrounding the mitochondria.”

→ “Protein crosslinking could be a possible way to increase viscosity in cells by inducing aggregation of mitochondrial or mitochondria-contacting organelle proteins (i.e., proteins in the mitochondrial matrix, ER membrane, and peroxisome)”

Q16: How elevated are the levels of the proteins after transfection compared to their normal levels?

Does transfection change the potency of the photosensitiser?

Response:

The expression level of proteins with EGFP (Sec61b, PEX16, and PTBP1) can be different from the expression level in normal condition. However, it is not important in this manuscript because we only utilised the proteins to control target location of EGFP for measuring crosslinking efficiency depending on cellular organelles, not to use their biological functions.

The primary factor that could change the potency of photosensitiser by transfection of EGFP might be energy transfer from EGFP to the **Ir-OA**. However, there is little possibility because the absorption wavelength region of the **Ir-OA** is not matched well to the emission region of the EGFP.

Q17: L202

“Protein oxidation by photoactivated Ir-202

OA occurred in the mitochondria, ER, and peroxisome”

Not in the cytoplasm – not even in those proteins surrounding the mitochondria?

(as suggested earlier)

Response:

We appreciate your insightful comment. As mentioned above, the photoactivation effects (including methionine oxidation) are on the mitochondrial membrane. To investigate the oxidation effect of the **Ir-OA** on the proteins of cytoplasm, we analysed three kinds of ratio based on our proteomics data, (i) significantly oxidised proteins of cytoplasm ($\log_2(\text{Fold change}) > 0$ and $-\log P$ value > 1) : whole oxidised proteins of cytoplasm = 123 : 2442 (5.0%), (ii) significantly oxidised proteins of mitochondria and cytoplasm ($\log_2(\text{Fold change}) > 0$ and $-\log P$ value > 1) : whole oxidised proteins of mitochondria and cytoplasm = 24: 349 (6.9%), and (iii) the significantly oxidised cytoplasmic proteins ($\log_2(\text{Fold change}) > 0$ and $-\log P$ value > 1) that are discovered as exists around OMM (Hung et al. eLife, **6**, e24463 (2017)) : the proteins that are discovered in the same reference = 5 : 34 (14.7%). This result is provided in the Supplementary data Set (O-Met proteome.xlsx). Interestingly, the ratio of oxidised cytoplasmic proteins got increase as the spatial extent is limited toward to the mitochondrial membrane (i < ii < iii), implying that the cytoplasmic proteins around mitochondria can be oxidised by the photoactivation of the **Ir-OA**. The analysis of cytoplasmic proteins was added on our Supplementary data spreadsheet (sheet 6 and 7), and we revised the manuscript as below.

Page 11 Line 248

“Protein oxidation by photoactivated **Ir-OA** occurred in the mitochondria, ER, and peroxisome, closely corresponding with the crosslinking efficiency results (Figure 5D).”

→ “Protein oxidation by photoactivated **Ir-OA** mainly occurred in the mitochondria, ER, and vesicles, and proteins in the cytoplasm and nucleus were partially oxidised as well, which closely corresponded to the crosslinking efficiency results (Figure 5D).”

Q18: L227

“were provided with an indication of the O-Met site (“

Meaning?

Response:

For clear understanding, we revised the sentence like below:

Page 12 Line 274

“Note that the crystal structures of the voltage-dependent anion channel (VDAC1), OXPHOS complex I matrix arm (NDUFS1 and NDUFA9), and OXPHOS complex III (CYC1 and UQCRC1) were provided with an indication of (Figure 5F)”

→ “Note that the crystal structures of the voltage-dependent anion channel (VDAC1), OXPHOS complex I matrix arm (NDUFS1 and NDUFA9), and OXPHOS complex III (CYC1 and UQCRC1) were described to indicate the O-Met site (Figure 5F)”

Q19: L229

“proteases oxidation occurs”

protease oxidation occurs

Response:

Including the typo, we rearranged the sentence for logical flow and clear understanding considering the position of referred Figure 6B.

Page 12 Line 278

“Additionally, proteases oxidation occurs, which implies that the dysfunction of mitochondrial proteins is not properly restored” (removed)

Page 13 Line 288

→ “In our O-Met proteome, several protease and chaperones were oxidised (Figure 6C), which could damage their functions of eliminating and restoring un-/misfolded mitochondrial proteins. Therefore, it accumulated damaged proteins and triggered corresponding mitochondrial fission/fusion and swelling” (Added)

Q20: L230

“is not properly restored”

After what? cell division?

Response:

That sentence means that the misfunctioned proteins could not be restored or eliminated by oxidised proteases. Therefore the sentence was revised as above (Response of Q19).

Q21: L230

“This oxidative stress for each crucial protein accelerates oxidation-induced cell death”

Perhaps

Response:

Page 12 Line 280

“accelerates”

→ “could be related to the acceleration of”

Q22:

“light irradiation”

Details of irradiation should be mentioned in the text

Response:

All sentence including ‘light irradiation’ was revised to mention details (Light source and irradiation energy).

Q23: L257

“ROS oxidises proteins”

ROS oxidises proteins

Q24: L266

“we reported”

we report

Response:

Page 14 Line 324

“reported” → “report”

Q25: P288

How long were the cells irradiated for?

Response:

Page 15 Line 350

We added the irradiation time as “3.75 mW/cm² for 1 min”

Q26: P332

“acetyl of the n-terminus”

acetyl of the N-terminus

Response:

Page 17 Line 395

“n-” → “N-”

Q27: L393

Ref 22 not properly formatted

Response:

The format of reference 22 was properly revised.

Q28: L490

“Localisation matter of Ir-OA in living cells”

Rephrase

Response:

Page 23 Line 577

“Localisation matter of **Ir-OA** in living cells”

→ “**Localisation of Ir-OA** in living cells”

Q29: L494

Figure 2C seems to show Ir-OA spanning a width of 1.5 um (1,500 nm). This is much bigger than a mitochondrion (ca. 300 nm) so it is not clear what is being observed.

What is missing are dark and light IC50 values and an indication of how many cells are damaged under the conditions of the confocal imaging.

Response:

As you point out, the mitochondria of Figure 2B and 2C are bigger than normal mitochondria. This is because a confocal laser rapidly photoactivates the **Ir-OA** and swells mitochondria. Therefore, we had tried to re-image the detailed localisation of the **Ir-OA** in the mitochondria with minimal laser power. However, it was impossible to distinguish normal mitochondrial (300 ~ 500 nm) membrane and matrix, although we utilized the super-resolution microscopy technique (Airyscan) (Figure R5). Therefore, we needed to image swelled mitochondria (1 ~ 1.2 μm) to resolve the mitochondrial membrane and matrix for identifying the specific location of **Ir-OA**.

To swell mitochondria, we scanned mitochondria with a confocal laser before imaging. The laser power ($\lambda = 405 \text{ nm}$) was 0.0125 mW, which is enough to damage mitochondria (*e.g.* the left cell of revised Figure 2B are scanned by a confocal laser before imaged, and the irradiation energy was 0.100 J cm^{-2} that is calculated by image size, $21.96 \mu\text{m} \times 21.96 \mu\text{m}$, and scanning time, 3.87 s). Considering that the 4 μM of the **Ir-OA** could damage mitochondria under irradiation of 0.17 J cm^{-2} (Supplementary Figure 11), the irradiation energy is enough to swell mitochondria. By using this technique, we re-imaged detailed localisation of the **Ir-OA** and revised Figure 2 and corresponding legend as below:

Figure R5. Co-localisation image of the **Ir-OA** with MitoTracker targeting mitochondrial membrane (A, B) and Mito-EGFP targeting mitochondrial matrix (C); A line-cut signal intensity of region of interest (ROI) is described right. (A) Co-localisation image after mitochondrial swelling by photoactivation of the **Ir-OA** (left). The mitochondrial matrix, cristae, and membrane can be identified, and the **Ir-OA** signal is observed on the mitochondrial membrane. (B) Co-localisation image using MitoTracker without mitochondrial swelling. (C) Co-localisation image using Mito-EGFP without mitochondrial swelling. The mitochondrial matrix and membrane are not identified in the image obtained without swelling.

Figure 2. Localisation of **Ir-OA** in living cells. A) Confocal images of **Ir-OA** with MitoTracker. Phosphorescence of **Ir-OA** (red) and fluorescence of MitoTracker (green) and merged image ($\lambda_{\text{ex}} = 405 \text{ nm}$ for **Ir-OA**, $\lambda_{\text{ex}} = 647 \text{ nm}$ for MitoTracker® Deep Red FM). Pearson's coefficient was calculated using Image J software ($R^2=0.86$). B) Airyscan confocal image of mitochondria with **Ir-OA** (red) and MitoTracker (green, outer/inner mitochondria membrane) for identifying the specific location of **Ir-OA**. C) Airyscan confocal image of mitochondria with **Ir-OA** (red) and Mito-EGFP (green, mitochondrial matrix). Line profiling for clarifying the specific location of each emission signal was followed by corresponding images (right). The **Ir-OA** signal is well merged with the MitoTracker signal on the outer/inner mitochondrial membrane and enclose Mito-EGFP signal of the mitochondrial matrix. The mitochondria of Airyscan images were pre-swelled by photoactivation ($\lambda_{\text{ex}}=405 \text{ nm}$, 0.0125 mW) of the **Ir-OA** for identifying mitochondrial substructure (the mitochondrial swelling effect of the **Ir-OA** is explained later). Line profiling analysis was proceeded with Carl Zeiss ZEN 3.0 Blue software. Conditions: [**Ir-OA**] = $4 \mu\text{M}$, [MitoTracker® Deep Red] = 100 nM , incubation time = 2 h and 0.5 h , respectively. Source data are provided as a Source Data file.

Q30: L115: “This super-resolution imaging result supports that the sub-mitochondrial localisation of Ir-OA might be from IMM spreading to the IMS and outer mitochondrial membrane (OMM) (Figure 2C).”

In several places protein cross-linking is shown as coupled tyrosines – what is the evidence for that in the present work?

Response:

As mentioned above, we tried to detect dityrosine crosslinking inside cells by using a dityrosine antibody. However, we could not conclude that the dityrosine bonds are a dominant part of protein crosslinking. Therefore, we removed figures describing dityrosine crosslinking (Detailed response is explained in Q3).

Q31: SI

L155

“Pd(II)OAc”

Pd(II)(OAc)₂?

Response:

“Pd(II)OAc” → “Pd(II)(OAc)₂”

Q32: Ir-OA: CHN analysis? HPLC purity?

Response:

We checked the HPLC purity of the **Ir-OA**, and the HPLC purity of the **Ir-OA** was 96.5% (Figure R6). So we added HPLC purity to the Supplementary Information 3-(9) section. However, we did not need elemental analysis because high-resolution mass spectrometry (HRMS) analysis was already provided (HR-MS: m/z calculated for C₇₂H₆₆IrN₈O₃: 1283.4894 (M⁺), found: 1283.4888 (M⁺)). The HRMS is much more accurate for identifying compounds than elemental analysis because the organic solvent or water molecules cause measurement error during the process of elemental analysis, which make it difficult to obtain consistent and accurate results.

Figure R6. HPLC purity of **Ir-OA**. The integrated area for the **Ir-OA** (7.5 min ~ 8.5 min) is 11.53722, and that for the impurities (2.0 min ~ 3.5 min) is 0.41993. The HPLC purity of the **Ir-OA** was calculated as 96.5% (The integrated area for the **Ir-OA**/total area).

Q33: Cell culture

Needs to be stated how the compound was added to the cells. Dissolved in DMSO?

Response:

All Iridium(III) compounds were dissolved in DMSO, then added to cell culture media in a volume ratio of 0.1% (Culture media:DMSO = 1000:1, v/v). So, we inserted a sentence, '(Culture media:DMSO = 1000:1, v/v)', to every part of the supplementary information that mention about it.

Q34: The switch between HeLa, A549 and HEK cells needs to be explained. I did not see A459 mentioned at all in the text (just SI). It cannot be assumed that different cell types will respond in the same way.

Response:

Really appreciate for reviewer's incisive inquiry. For photo-crosslinking analysis, we repeated experiment with HeLa cells and the results showed that photo-crosslinking was similarly presented like photo-crosslinking result of HEK cells. Therefore, protein modification by **Ir-OA** was also generated in HeLa cells.

In the view of mass spectrometry (MS) analysis, HEK cells are the most frequently utilized model system. Actually, mass spectrometry proteomic mapping experiment has been accomplished with HEK cells. In this point, we chose the HEK cells for mass spectrometry analysis. In previously reported whole proteome analysis depending on cell lines, the correlation value between HEK cells and HeLa cells is high ($r = \sim 0.8$) compared to other cancer cells (U2OS; bone osteosarcoma, A549; lung adenocarcinoma, RKO; colon carcinoma, and MCF-7; breast cancer cell), indicating that protein pool is almost similar to each other (*Mol. Cell. Proteomics* **11**, 1-11 (2012)). In addition, we removed MTT assay experiments with A549 cells because this experiment was not logically connected to the rest parts of the manuscript.

Q35: Basic IC_{50} data on them all are missing.

Response:

We added a supplementary table for IC_{50} corresponding to irradiation energy. The IC_{50} value was calculated based on CCK-8 assay as described in Q10 (Supplementary Table 2).

Q36: Are any of the Ns protonated at pH 7 in the side arm of Ir-OA?

Response:

There are three Ns where protonation is possible: N of benzoxazole, N of amides, and N of tertiary amine. The pKa of these conjugate acids of Ns are reported as 0.18 (benzoxazole), -0.5 (amides), and 4.83 (N,N-Dimethyl-1-naphthylamine) respectively. In the previously reported literature (Kim et al. *Angew. Chem. Int. Ed.* **47**, 2231–2234 (2008)), the AH1 and AH2 having similar molecular structure to energy donor of the **Ir-OA**, are protonated at pH under 6. Thus, there is no protonated Ns in physiological conditions.

Reviewer #2 (Remarks to the Author):

This manuscript reports an iridium complex that can produce ROS upon irradiation, and at the same time report on viscosity changes in the immediate environment. Importantly, a negative control analogue is used throughout the paper. The authors have used this complex in detailed biological studies of cell death caused by mitochondrial oxidative stress. This work encompasses a range of elegant studies across chemistry and biology, and is therefore appropriate for publication in this journal, after the following amendments:

Q1: While the negative control does provide some information, the authors also need to include irradiation alone as another negative control in their studies. It has been reported that 400 nm light does induce ROS, so studies should be carried out to ascertain whether this contributes significantly to the results reported here. This is particularly important for the use of DCFDA: this is now largely rejected as a meaningful reporter of ROS, and can be photoactivated itself, so the control experiment is required here.

Response:

We would like to thank to this helpful comment. As you pointed out, we had to include negative controls (+Ir-OA/-hv, +Ir-OC/-hv, and +hv only) to the H₂DCF-DA assay. Thus, we re-conducted the experiment, and found that DCF signal was enhanced in the condition of +IrOA/+hv while the other negative controls showed no DCF signal (Supplementary Figure 11). Furthermore, we confirmed that the H₂DCF-DA was not oxidised by 400 nm light irradiation only because of very weak light irradiation intensity in our experimental condition (0.17 J cm⁻²). In contrast, we found that the H₂DCF-DA was activated by light irradiation for 10–20 min by solar simulator (60–120 J cm⁻²).

Supplementary Figure S11. H₂DCF-DA assay for identification of ROS generation inside live cells. The green fluorescence of DCF turned on after photoactivation of **Ir-OA** or **Ir-OC**. Conditions: [**Ir-OA** or **Ir-OC**] = 4 μM, [H₂DCF-DA] = 20 μM, 400 nm light LED array (0.170 J cm⁻²), control: 400 nm light irradiated HeLa cells without incubation of iridium complexes.

Q2: Both IrOA and IrOC are referred to as photosensitisers throughout, but IrOC in fact cannot act as a photosensitiser – that is why it is the negative control. The terminology should therefore be amended.

Response:

Thanks to your comments. We revised the word ‘photosensitisers’ to the ‘**iridium(III) complexes**’ when the ‘photosensitisers’ means both of **Ir-OA** and **Ir-OC**.

Q3: The live-dead assay described here does not provide such valuable information as no statistical data is provided, and the red and green signals are not correlated. This assay should be repeated by flow cytometry to provide these data. Furthermore, an apoptosis-necrosis assay could also be done at the same time to confirm the hypothesis that photoactivation of IrOA induces apoptosis.

Response:

Considering your constructive comment, we repeated a live/dead assay using fluorescence-activated cell sorting (FACS). The results were described as the 2D histogram of PI/Calcein AM (for live/dead assay) and PI/Annexin V (for confirming apoptotic cell death) (revised Figure 2). Under weak light irradiation of 0.25 J cm⁻², most of HeLa cells with **Ir-OA** show

positive PI/negative Calcein AM signal, besides even the cells of negative PI cells show positive Annexin V signal. These results imply that the photoactivation of **Ir-OA** strongly triggers apoptosis even under weak light. Thus, we added corresponding figures and sentences to the manuscript as below.

Page 7 Line 145

→ “The live/dead assay was confirmed by fluorescence activated cell sorting (FACS) and assessed using 2D histograms (Figure 3C and Supplementary Figure 15). When **Ir-OA** was photoactivated, the number of HeLa cells with positive PI and negative Calcein AM dramatically increased, confirming that most of the cells were dead. In addition, the number of cells with positive Annexin V and negative PI (Q2) increased upon photoactivation with **Ir-OA** prior to the disruption of the cell membrane (positive PI) (Figure 3D and Supplementary Figure 15), showing that the dying cells undergo an early apoptosis stage. Hence, the photoactivation of **Ir-OA** triggered apoptotic cell death while that of **Ir-OC** did not alter cell viability” (added)

Figure 3. Identification of phototoxicity effect from iridium(III) complexes. A) CCK-8 assay for quantifying the cytotoxicity of **Ir-OA** and **Ir-OC** with or without light irradiation for HeLa cells. All error bars = s.d. (n=3). Conditions: photosensitiser iridium(III) complexes incubation time = 2 h, light source = 400 nm light LED array, light dose: 0.08 J cm⁻², 0.17 J cm⁻², and 0.25 J cm⁻². B) Live/dead assay for verifying phototoxicity of **Ir-OA** and **Ir-OC**. At 6 and 20 hours after light irradiation, dead and live cells were stained using propidium iodide (PI, red) and Calcein AM (green), respectively (scale bars = 200 µm). Conditions: [Iridium(III) complex] = 8 µM, light source = 400 nm light LED array, light dose = 0.255 J cm⁻². C) Representative Calcein AM vs PI flow cytometry plot for HeLa cells incubated with iridium complexes. D) Representative Annexin V vs PI flow cytometry plot for HeLa cells incubated with iridium complexes. Q1 (right top): late apoptotic/necrotic cells, Q2 (right bottom): early apoptotic cells, Q3 (left bottom): viable cells, and Q4 (left top): necrotic cells. Conditions for flow cytometry: [Ir complex] = 8 µM, light source = 400 nm light LED array, light dose = 0.25 J cm⁻². Source data are provided as a Source Data file.

Q4: For both the FLIM studies in cells, and the BSA studies in cuvette, statistical information must be provided, otherwise it is impossible to ascertain whether these results are significant. This is particularly important for the cells, where a change from 844 to 911 ns could easily not be significant.

Response:

We are sincerely sorry for not providing statistical information of FLIM and BSA studies. Firstly, we triplicated the lifetime imaging under the same condition (w/ and w/o light irradiation) (Figure R7), and the FLIM images exhibited similar averaged lifetime change. Thus, we supposed that there was significant change in mitochondrial viscosity and revised the manuscript as below.

Page 8 Line 175

→ “HeLa cells with **Ir-OA** exhibited an average lifetime of 866 ± 20 ns before light irradiation (LED array, $\lambda = 400$ nm, 0.17 J cm^{-2}), which increased to 915 ± 14 ns following irradiation, likely due to the accumulation of crosslinked proteins

Figure R7. FLIM images of HeLa cells with **Ir-OA** after light irradiation (A) and without light irradiation (B).

Second, we found that we made a mistake in measuring lifetime of **Ir-OA** in BSA solution. When we measure the lifetime of dark condition, the photon counting baseline was not corrected for unknown reasons (Figure R8), which led to a misconception that the lifetime of **Ir-OA** seemed to be abnormally short. We repeated the experiment with corrected baseline, but there was no significant change in the lifetime of **Ir-OA** because the lifetime of 1900~2100 ns is almost saturated value, and **Ir-OA** could not efficiently crosslink BSA (Figure R3). Thus, we reconstructed the BSA experiment; we measured lifetime of **Ir-OA** depending on the BSA concentration to estimate how much the lifetime of **Ir-OA** varies with the local concentration of protein around mitochondria. The result showed that the lifetime of **Ir-OA** increased according to the BSA concentration (from 664 ± 31 ns at 0.0156 mg/ml to 1912 ± 40 ns at 4.00 mg/ml) (Supplementary Figure 16), implying that protein accumulation by crosslinking and corresponding viscosity increase can be detected by change in lifetime of **Ir-OA**. So, we revised the manuscript as below.

Page 8 Line 166

→ “In addition, we confirmed that the increased protein concentration of local area causes the phosphorescence lifetime enhancement of **Ir-OA**. The Bovine serum albumin (BSA), known to increase the viscosity of solution in according to its concentration³⁵, was dissolved in aqueous solution of **Ir-OA** at concentration of 0.0156, 0.0625, 0.250, 1.00, and 4.00 mg/mL (Supplementary Figure 17). Then, the phosphorescence lifetime of **Ir-OA** was measured, resulting in an increase from 664 ± 31 ns to 1912 ± 40 ns as the concentration of BSA increases. This result implies that local accumulation of proteins by photocrosslinking and corresponding viscosity increase can be monitored by change of the phosphorescence lifetime of **Ir-OA**” (Added)

Page 8 Line 177

~~→ “Bovine serum albumin (BSA), known to be crosslinked by photoactivation of photosensitisers³⁵, was utilised to verify the relationship between protein crosslinking by photoactivation of **Ir-OA** and phosphorescence lifetime. **Ir-OA** was dissolved in BSA solution; the phosphorescence lifetime of **Ir-OA** was observed to dramatically increase from 646 ns (before irradiation) to 1,735 ns (after irradiation), implying that crosslinked BSA triggered enhanced viscosity (Figure 4D).” (Removed)~~

Figure R8. The raw data of TCSPC for measuring lifetime of **Ir-OA** according to BSA crosslinking. The baseline of dark condition was not corrected (right). All other TCSPC results have corrected baseline.

Supplementary Figure S16. The change in the lifetime of **Ir-OA** according to BSA concentration. The lifetime of **Ir-OA** was measured by TCSPC at each BSA concentration of 0.0156, 0.0625, 0.250, 1.000, and 4.000 mg/mL. condition: [**Ir-OA**] = 20 μ M, λ_{ex} = 450 nm, n=3.

Q5: There is clearly some complex that is not in the mitochondria, and the authors conclude some effect of the complex on the ER and peroxisomes, among all organelles, but there is no evidence for this. Since the GFP-labelled organelles are available, co-localisation studies should be carried out of IrOA with all 4 of these labelled cells.

Response:

We really appreciate this valuable comment. We carried out co-localisation studies of **Ir-OA** with four kinds of labelled HeLa or HEK293T cells. As shown in the revised Figure 4C-4F and Supplementary Figure 19, the labelled HEK293T cells and HeLa cells are imaged with **Ir-OA**. Interestingly, the Pearson's coefficients order (Mitochondria > ER > peroxisome > Nucleus) was similar to the order of crosslinking efficiency, representing that the proteins of organelles contacting mitochondrial membrane can be crosslinked by photoactivation of **Ir-OA**. Notably, the crosslinking efficiency of mitochondrial proteins was less than that of ER proteins (and peroxisome proteins in HeLa cells), which might be because the Mito-EGFP was expressed in the mitochondrial matrix, while **Ir-OA** is located on OMM and IMM. Thus, the **Ir-OA** molecules on the outer face of IMM and OMM exhibited relatively more crosslinking ER membrane proteins than mitochondrial matrix.

Page 9 Line 196

→ “The crosslinking efficiency difference is affected by the possibility of contact between **Ir-OA** and EGFP of each organelle. Therefore, we transfected four different EGFP constructs again and imaged the EGFP with **Ir-OA** to investigate their proximity (Figures 4C-F, right). The Pearson's coefficients R of each image were then calculated (Mito-EGFP, $R = 0.907$; Sec61b-EGFP, $R = 0.477$; PEX15-EGFP, $R = 0.124$; PTBP1-EGFP, $R = -0.347$ vs **Ir-OA**). Considering that the EGFP closer to **Ir-OA** is expected to be more easily crosslinked, it can be reasoned that the EGFP in the mitochondrial matrix, ER membrane, and peroxisome were more crosslinked than the EGFP in the nucleus. Notably, the crosslinking efficiency of ER membrane proteins was more significant than that of mitochondrial matrix proteins and peroxisome proteins because **Ir-OA** is located on the outer surface of the OMM and IMM. The ER membrane proteins were in direct contact with **Ir-OA**, while the contact between the proteins in the mitochondrial matrix and **Ir-OA** is limited by the IMM. Therefore, the **Ir-OA** molecules triggered relatively more protein crosslinking in the ER membrane than that in the mitochondrial matrix.” (added)

Figure 4. Mitochondrial viscosity changes with photo-crosslinking using photoactivation of **Ir-OA**. A) Viscosity-dependent change in the lifetime of **Ir-OA**. The in vitro viscosity was precisely controlled with glycerol content change in MeOH (v/v, %) from 0% to 95%. $n=3$. B) Phosphorescence lifetime image (PLIM) of **Ir-OA** in mitochondria before and after photoirradiation. The described lifetime is averaged value of three images obtained under same condition. ($n=3$) C-F) Western blot (left) for identification of protein photo-crosslinking of **Ir-OA** depending on four different cell organelles: C; mitochondria (Mito-EGFP), D; ER (Sec61B-EGFP), E; peroxisome (PEX16-EGFP), and F; nucleus (PTBP1-EGFP). Co-localization images (right) of **Ir-OA** (red signal) and each EGFP (green signal). $\lambda_{ex} = 405$ nm and 488 nm (**Ir-OA** and EGFP, respectively). Pearson's coefficient for respective cell organelles with **Ir-OA** was calculated using Image J software (Mito-EGFP, $R=0.907$; Sec61b-EGFP, $R=0.477$; PEX15-EGFP, $R=0.124$; PTBP1-EGFP, $R=-0.347$ vs **Ir-OA**). Line-cut analysis (bottom) of western blot signals with or without photo-irradiation was performed to quantify the crosslinking efficiency (η) ($n=3$). Each correlation value (R^2) indicating similarity was written above the line cut spectrum. Note that higher efficiency for ER than mitochondria is for significant OMM/IMM location of **Ir-OA**. Source data are provided as a Source Data file. All errors = s.d. ($n=3$).

Supplementary Figure S19. Protein photo-crosslinking by photoactivation of **Ir-OA** in HeLa cells. Western blot (left) for identification of protein photo-crosslinking of **Ir-OA** in HeLa cells depending on four different cell organelles: mitochondria (Mito-EGFP), ER (Sec61B-EGFP), peroxisome (PEX16-EGFP), and nucleus (PTBP1-EGFP). Note that transfection with PEX16-EGFP construct in HeLa is insufficient, thereby, the signal on the lane after photo-crosslinking reaction does not seem to appear. Co-localization images (right) of **Ir-OA** (red signal) and each EGFP (green signal). $\lambda_{\text{ex}} = 405 \text{ nm}$ and 488 nm (**Ir-OA** and EGFP, respectively). Pearson's coefficient for respective cell organelles with **Ir-OA** was calculated using Image J software (Mito-EGFP, $R = 0.891$; Sec61b-EGFP, $R = 0.634$; PEX16-EGFP, $R = 0.181$; PTBP1-EGFP, $R = 0.055$ vs **Ir-OA**). Imaging conditions: [**Ir-OA**] = $4 \mu\text{M}$, $\lambda_{\text{ex}} = 405 \text{ nm}$ for **Ir-OA**, $\lambda_{\text{ex}} = 488 \text{ nm}$ for EGFP. Emission gain: $550\text{--}650 \text{ nm}$ for **Ir-OA** and $500\text{--}550 \text{ nm}$ for EGFP. Line-cut analysis (bottom) of Western blot signals with or without photo-irradiation was performed to quantify the crosslinking efficiency (η). Each correlation value (R^2) indicating similarity was written above the line cut spectrum. Source data are provided as a Source Data file.

Q6. Figure S18 – the figure caption must specify what the merged images correspond to.

Response:

We revised the figure legend to specify what the merged images means. The merged images and their colour were obtained by combining each wavelength image. Thus, we correct the legend like below:

Supplementary Figure 21. Wavelength resolved CLSM images of mitochondrial depolarisation. Lambda scanning images was obtained from 424 nm to 620 nm at the 18 nm interval before/after irradiation. Merged images and their colour (left) describe overlapping images gained at each wavelength. Yellow region of merged image (left top) corresponds to polar mitochondria (there is no signal at blue region), and the blue region of merged image (left bottom) corresponds to depolarised mitochondria. Conditions: [Ir-OA] = 4 μ M, light source = 400 nm light LED array (0.170 J cm⁻²).

Reviewer #3 (Remarks to the Author):

This paper describes strategy based to monitor intramolecular energy transfer to produce mitochondrial oxidation-induced cell death. The synthesis of the reagent and all supporting analyses is well-documented in the supporting information.

Various imaging techniques were then employed to monitor the mitochondrial responses to this strong oxidative stress. Microviscosity and depolarisation were caused by protein crosslinking around the mitochondria due to the application of this reagent. Oxidised methionine residues were used to report upon the mitochondrial proteins that were affected by the process and a label free proteomics approach was adopted to tabulate these results. Many of the observations were interpreted in terms of effects associated with the channel and translocase proteins as well as the OXPHOS chain.

The experiments are on the whole well-designed and well-executed and I recommend publication following attention to some minor issues:

Points to address:

Q1: Figure 5 e and a number of the other figures are hard to follow – primarily since the font used for labelling is so small and the details are not well described in the legends.

Response:

Thanks to your comments. We apologize that the font size of some figure is so small, which make it difficult to follow contents. Accordingly, we raised the font size of Figure 5E and complemented the legends to describe what the figures represent as below. In addition, we revised the other legends that did not fully describe figures and increased the font size of Figure 1 and 2.

Figure 5. Mitochondrial depolarisation by photoactivation of **Ir-OA** and related proteomics.

A) Schematic illustrating polarity dependent energy transfer efficiency changes and following ratiometric emission property changes depending on the H₂O:MeOH ratio. ~~B) Ratiometric time lapse imaging in HeLa cells with Ir-OA according to oxidative stress (photoactivation) using CLSM. Ratiometric emission was observed 90 min following oxidative stress exposure (LED array, $\lambda = 400$ nm; light dose = 0.17 J/cm²) (top, scale bars = 50 μ m) or during photoactivation within 60 s in real time (bottom, scale bars = 20 μ m). The CLSM instrument's laser excited Ir-OA for real-time imaging (bottom). Ratio = (emission of acceptor, $\lambda_{em} = 573$ – 620 nm/emission of donor, $\lambda_{em} = 420$ – 480 nm). Normal mitochondria with high MMP: red (top) and green (bottom).~~ B) Ratiometric CLSM imaging of HeLa cells with **Ir-OA** according to giving oxidative stress. Ratiometric emission was observed 0, 10, 45, and 90 min after oxidative stress exposure (LED array, $\lambda = 400$ nm; light dose = 0.17 J cm⁻²) (top, scale bars = 50 μ m), and the mitochondrial polarity change was monitored by the real-time ratiometric imaging during photoactivation within 60 s (bottom, scale bars = 20 μ m). The CLSM instrument's laser excited **Ir-OA** for real-time imaging (bottom). Ratio = (emission of acceptor, $\lambda_{em} = 573$ – 620 nm/emission of donor, $\lambda_{em} = 420$ – 480 nm). Normal mitochondria with high MMP: red (top) and green (bottom). C) Quantitatively analysed Volcano plot of oxidised proteome. The substantially oxidised proteome was sorted in the range of $\log_2(\text{Fold change}) > 0$ and $-\log P$ value > 1.0 (light blue region). D) Proportion of the oxidised proteome in various organelles. Proteome location was determined using UniProt and mitochondria proteins were cross-checked with Human MitoCarta2.0 dataset consisting of 1158 human genes. The proportion (%) was obtained as ratio between the number of significantly oxidized proteome ($-\log P > 1$) and that of whole oxidized proteome ($-\log P > 0$) in each cell organelle. E) Heat map with label free quantification (LFQ) values for 28 mitochondrial proteins among 112 substantially oxidised mitochondrial proteins (Supplementary Data Set). F) Three representative oxidised proteins; OXPHOS complex I (NDUFS1 and NDUFA9), OXPHOS complex III, and voltage-dependent anion-selective channel 1 (VDAC1). The crystal structures of three proteins from RCSB protein data bank (PDB ID) were visualised and processed with PyMOL. Source data are provided as a Source Data file.

Q2. It is interesting that the proteins for which effects were observed largely correlated with those that are present at the highest abundance.

Response:

We really thank for your comments on the interesting points of our manuscript.

Q3. Figure S20 to me made the paper more comprehensible – I would suggest promoting this to the introduction or conclusion of the paper.

Response:

We really appreciate your suggestion. We changed the Figure S20 to main Figure 6E and added corresponding text to conclusion part as below:

Page 13 Line 288

→ “In our O-Met proteome, several protease and chaperones were oxidised (Figure 6C), which could damage their functions of eliminating and restoring un-/misfolded mitochondrial proteins. Therefore, it accumulated damaged proteins and triggered corresponding mitochondrial fission/fusion and swelling. Fission/fusion, known as the protein quality control process, could be related to oxidation of mitochondria/ER proteases and chaperones. Thus, fission/fusion is overloaded due to the accumulation of damaged proteins inside the mitochondria caused by the dysfunction of proteolysis. The phenomenon corresponds to the fact that the proteins involved in fission and fusion were not observed in the O-Met proteome (Figure 6D). Further, mitochondrial matrix swelling could be explained by the oxidation of OXPHOS and channel proteins leading to an ion imbalance (Figure 6E).” (added)

Figure 6. Mitochondria morphology monitoring and proposed cell death mechanism. A) Time-lapse Airyscan 2 images of HeLa cells with Ir-OA before light irradiation (14 mW, 405 nm laser of laser scanning microscopy) (left) and after irradiation for 204 s (right). To monitor morphological changes, the mitochondrial matrix was transfected by Mito-EGFP (green signal) (scale bars = 10 μ m). White boxes indicate mitochondrial fission and fusion. B) Enlarged time lapse images (0–340 s) of white boxes from Figure 6A. Along with fission and fusion mitochondria swelling was also observed C) Investigation of proteases and mitochondrial fission and fusion-related protein oxidation. Heat map diagram for O-Met proteins related to mitochondrial protein quality control, in addition to D) fission, and fusion. E) Brief description of a mechanism for mitochondrial environment change. This illustrates the impact of mitochondrial oxidative stress based on proteomic analysis of mitochondrial oxidative stress. F) Mechanistic description of mitochondrial oxidation-induced cell death based on phenomenological observations and proteome analyses.

Q4. I think the concluding remarks oversell the paper too strongly – this a good well-executed study that I recommend for publication but am not sure that it will go far in solving problems in mitochondrial diseases – this link was not apparent at least to me.

We really appreciate your remark and agree that some part of our conclusion was exaggerated. To make our conclusion acceptable, we revised the related sentence and paragraph in the introduction and conclusion sections like below.

Page 2 Line 19

“Mitochondrial oxidation-induced cell death, a physiological process implicated in ageing and the pathogenesis of various diseases, has been ~”

→ “Mitochondrial oxidation-induced cell death, a physiological process triggered by various cancer therapeutics to induce oxidative stress on tumours, has been ~”

Page 3 Line 40

“~ is essential to understanding cellular ageing and the pathogenesis of various diseases.”

→ “~ is essential to understanding and improving cancer therapeutics based on oxidative damage to tumours.”

Page 4 Line 66

“ ~ will pave the way to understanding cell ageing, death, and pathogenesis triggered by oxidative stress.”

→ “~ will aid the understanding of cancer therapeutics induced mitochondrial oxidative stress.”

Page 15 Line 335

“These results elucidate the pathology of mitochondrial oxidation-related diseases and provide a fundamental understanding of mitochondrial oxidation-induced ageing and death. We believe that this research will serve to overcome the hurdles currently restricting advancement in human mitochondria-related diseases.”

→ “These results suggest the way in which mitochondrial photosensitisation affects cellular survival. We hope that these results will aid in providing a fundamental understanding of mitochondrial oxidation-related diseases as well as cancer therapeutics inducing oxidative stress.”

REVIEWER COMMENTS

Reviewer #1 (Remarks to the Author):

My comments and details of the further points which need attention are below.

This paper has required a lot of effort by reviewers. The authors should acknowledge the critical comments of the reviewers and thank them for their help in improving the script.

Page 4 Line 66 "

MAY aid the understanding of cancer therapeutics induced mitochondrial oxidative stress.

Page 15 Line 335

"These results suggest A way in which

We hope that these results will CONTRIBUTE TO a

Q3

However, the enhancement was not significant, so we concluded that the dityrosine bond was involved in the photo-crosslinking, but it does not seem the dominant factor of the photo-crosslinking. Thus, we removed related figures (Figure 4A and 6C).

I note the removal of these data.

I am surprised there is no mention of tryptophan oxidation.

"most of them show uniform changes in their characteristic such as"

Where are the quantitative data to support this statement?

As they say

"it is also difficult to define mitochondrial viscosity because of its complex structure"

Hence I don't find it to be a useful concept.

"Notably, the phototoxicity of Ir-OA derived by CCK-8 assay is lower than that derived by MTT assay"

"Thus, we changed the cell viability data from the MTT assay to CCK-8 assay of Figure 3"

This change of assay is noted.

Table S2

Error bars are needed

Q11

"ROS and mitochondrial oxidative stress triggers microenvironment changes, which include increased cytoplasmic and mitochondrial viscosity"

This is surely meaningless in view of the heterogeneity of both the cytoplasm and mitochondria - depending on how you define the cytoplasm. About 30% of the cytoplasm is usually thought to be in membrane-bound organelles, rest is cytosol.

Fig. R3

"Note that in vitro photo-crosslinking is less efficient than in cell"

I don't understand. There is no BSA cross-linking in cells, and why should different proteins behave in the same way?

Fig. R4

On what data is this distribution based ?

Q15

"The formation of 'protein aggregates' include crosslinked and damaged (oxidised) proteins of the

mitochondrial matrix, membrane, IMS, ER membrane, peroxisome, and maybe cytoplasm.”
“All these aggregated proteins can induce a viscous environment surrounding mitochondria”
A complicated situation- the arguments are not very convincing.

Q17

“proteins in the cytoplasm and nucleus were partially oxidised as well,”
I note the additional data.

Fig R5/Fig 2

What does the x-axis have micromolar concentration units?
Label needed for y axis

Q32

“However, we did not need elemental analysis because high-resolution mass spectrometry (HRMS) analysis was already provided (HR-MS: m/z calculated for C₇₂H₆₆IrN₈O₃: 1283.4894 (M⁺), found: 1283.4888 (M⁺)). The HRMS is much more accurate for identifying compounds than elemental analysis because the organic solvent or water molecules cause measurement error during the process of elemental analysis, which make it difficult to obtain consistent and accurate results”

Not true.

MS is not a good test for purity- depends on the introduction of ions into the gas phase. Neutral molecules and non-volatile impurities are not detected.

Elemental analysis detects everything that is combustible and the presence of solvents can be accounted for (and often verified by other means)

The added HPLC is very useful but needs care with interpretation since extinction coefficients may be different for different peaks (4 peaks in the present case). The detection wavelength, solvent and elution conditions are needed.

What is the HPLC purity of IrOOC?

Q34

Noted that “we removed MTT assay experiments with A549 cells because this experiment was not logically connected to the rest parts of the manuscript”

Other comments

Fig 1A: why has a specific enantiomer been drawn? A comment needs to be added on the chirality of the complexes they have used (presumably racemic).

Fig 1C: absorbance does not have arbitrary units (a.u.). In fact it does not have units at all. (log to base 10 of incident/transmitted radiant flux)

Reviewer #2 (Remarks to the Author):

The authors have thoroughly addressed all the reviewer comments, and the manuscript that they have presented is much clearer, and likely to be very impactful.

Two minor points:

Page 3, line 43 – “Oxidative stress induced by photosensitisers causes chemical modifications of biomolecules, which include proteins, unsaturated lipids and DNA.” Is confusing as it suggests that proteins, unsaturated lipids and DNA are types of biomolecules (which is, of course, true, but not the intended point. “Oxidative stress induced by photosensitisers causes chemical modifications of biomolecules including proteins, unsaturated lipids and DNA” or “Oxidative stress induced by photosensitisers causes chemical modifications of biomolecules such as proteins, unsaturated lipids

and DNA" would be much clearer

Page 5, line 64 – "The proposed mechanism will aid the understanding of cancer therapeutics induced mitochondrial oxidative stress." Should be "The proposed mechanism will aid the understanding of how some cancer therapeutics can induce mitochondrial oxidative stress."

Reviewer #3 (Remarks to the Author):

I believe that the authors have addressed all the concerns that i raised during the initial review and I am happy to recommend publication of this manuscript in your journal.

We carefully checked all reviewer's comments several times again and provided related data and explanation that have been revised according to the comments and standard guidance of Nature Communications. All the comments were very constructive and gave us an opportunity that can further develop this manuscript. Responses for all comments in revision are following.

Reviewer #1 (Remarks to the Author):

My comments and details of the further points which need attention are below.

This paper has required a lot of effort by reviewers. The authors should acknowledge the critical comments of the reviewers and thank them for their help in improving the script.

→ Response: We appreciate your constructive and detailed reviews and agree that the reviews are enormously helpful to improve this manuscript. We carefully rechecked your reviews and gave additional experimental data and explanation as below.

Page 4 Line 66 “

MAY aid the understanding of cancer therapeutics induced mitochondrial oxidative stress.

Page 15 Line 335

**“These results suggest A way in which
We hope that these results will CONTRIBUTE TO a**

→ Response: We revised the sentences as you pointed out. We thank you for your careful comments.

Page 4 Line 64

The proposed mechanism will aid ~ → **The proposed mechanism may aid ~**

Page 14 Line 310

These results suggest the way in which ~ → **These results suggest a way in which ~**

Page 14 Line 311

We hope that these results will aid in providing a fundamental understanding → **We hope that these results will contribute to providing a fundamental understanding.**

Q3

However, the enhancement was not significant, so we concluded that the dityrosine bond was involved in the photo-crosslinking, but it does not seem the dominant factor of the photo-crosslinking. Thus, we removed related figures (Figure 4A and 6C).

I note the removal of these data.

I am surprised there is no mention of tryptophan oxidation.

→ Response: As you pointed out, the tryptophan oxidation is also a candidate of protein dysfunction and crosslinking even though the tryptophan is the rarest amino acid in proteins.

Various studies have reported that tryptophan oxidation by photosensitizer causes crosslinking with other amino acid residues. (M. Ehrenshaft, L.J. Deterding and R.P. Mason, *Free Radic. Biol. Med.*, **89**, 220–228 (2015)), (Ludvikova et al., *J. Org. Chem*, **83**, 10835–10844 (2018)). In this manuscript, however, we did not cover detailed chemical reaction for the plausible photo-crosslinking pathway. We are on the way to clarify the detailed chemical reaction and expect that specific amino acid interaction (including tryptophan) involved in photo-crosslinking will be elucidated soon.

**“most of them show uniform changes in their characteristic such as”
Where are the quantitative data to support this statement?**

→ Response: We apologize that there was a misleading statement in our previous response. What we intended was that mitochondria show a unidirectional characteristic change in average under the specific stress. We could not provide quantitative data as evidence of ‘uniform change’ because of the heterogeneity of mitochondria, but we showed the unidirectional and averaged characteristic change in response to oxidative stress through the FLIM and ratiometric imaging. Previously reported papers also provided cellular images to show unidirectional characteristic changes including other factors (pH and temperature) like our study (Lee et al. *J. Am. Soc. Chem.* 2014, 136, 40, 14136–14142)(Huang et al. *Anal. Chem.* 2018, 90, 23, 13953–13959).

As they say

**“it is also difficult to define mitochondrial viscosity because of its complex structure”
Hence I don’t find it to be a useful concept.**

→ Response: As you pointed out previously, the structural complexity of mitochondria makes it difficult to define ‘mitochondrial viscosity’. However, considering that the **Ir-OA** locates on the IMM and OMM, the viscosity change which is observed in this manuscript may be around IMM, OMM, IMS, and matrix. Even though we could not precisely specify where the viscosity change occurs, we think it can be meaningful because the viscosity change around the mitochondrial membrane may affect reaction kinetics of metabolites.

“Notably, the phototoxicity of Ir-OA derived by CCK-8 assay is lower than that derived by MTT assay”

“Thus, we changed the cell viability data from the MTT assay to CCK-8 assay of Figure 3”

This change of assay is noted.

→ Response: We really appreciate your incisive comment.

Table S2

Error bars are needed

→ Response: We revised the Table S2 to include error bar.

Supplementary Table 2. Quantitative phototoxicity of Ir-OA according to irradiation energy.

Irradiation energy	IC ₅₀		
	0.085 J/cm ²	0.170 J/cm ²	0.255 J/cm ²
MTT assay (n=4)	2.348 ± 0.162 μM	1.583 ± 0.003 μM	1.302 ± 0.052 μM
CCK-8 assay (n=4)	15.849 ± 0.370 μM	11.036 ± 0.610 μM	7.305 ± 0.758 μM

The IC₅₀ values are measured by mitochondrial function-dependent (MTT assay) and independent (CCK-8) assay. The assays were triplicate at various irradiation energy (0.085, 0.170, and 0.255 J cm⁻²)

Q11

“ROS and mitochondrial oxidative stress triggers microenvironment changes, which include increased cytoplasmic and mitochondrial viscosity”

This is surely meaningless in view of the heterogeneity of both the cytoplasm and mitochondria -depending on how you define the cytoplasm. About 30% of the cytoplasm is usually thought to be in membrane-bound organelles, rest is cytosol.

→ Response: We thank you for pointing it out. There was a misunderstanding about the concept of cytoplasm. Thus we revised the previous sentence as below considering the ref. 16 and 31. (16. Hao et al. *Chem. Sci.*, **10**, 1285-1293 (2019) reporting mitochondrial viscosity increase in response to oxidative stress and 31. Kuimova et al. *Nat. Chem.*, **1**, 69-73 (2009) reporting a viscosity increase of cytoplasm).

Page 7 Line 145

“ROS and mitochondrial oxidative stress triggers microenvironment changes, which include increased cytoplasmic and mitochondrial viscosity^{16,31} and mitochondrial depolarisation^{15,32}.”

→ “ROS and following oxidative stress triggers a change in microenvironment such as viscosity^{16,31} and polarity^{15,32} at the specific region affected by the stress.”

Fig. R3

“Note that in vitro photo-crosslinking is less efficient than in cell”

I don’t understand. There is no BSA cross-linking in cells, and why should different proteins behave in the same way?

→ Response: Photo-crosslinking by the catalytic reaction is potentially generated on the amino acid residues on tyrosine, tryptophan, histidine, and lysine, which are the component of proteins. The kind of protein is not important. Therefore, we chose the BSA protein accessible easily.

The reason that in vitro photo-crosslinking is less efficient than in-cell experiments would be the average distance between **Ir-OA** and proteins. In-cell experiments, the **Ir-OA** seems to be fixed on the membrane, which can increase the reaction rate because of the close distance between **Ir-OA** and surrounding proteins, compared to diffusive *in vitro* experimental condition. Additionally, various electron donating biomolecules in cells may promote electron transfer of **Ir-OA**. To further confirm our claim, we performed the BSA crosslinking experiment again under the higher concentration of **Ir-OA** (100 μM) and stronger light intensity ($\lambda_{ex}=405$ nm, 0.71 mW/cm² for 0, 10, 30, and 60 min). As a result, we significantly confirmed that the **Ir-OA** induces photo-crosslinking of BSA (Figure R1).

Lane	#1	#2	#3	#4	#5
100 μ M Ir-OA	-	+	+	+	+
Irradiation time (min)	60	0	10	30	60
BSA	+	+	+	+	+

Figure R1. In vitro bovine serum albumin (BSA) photo-crosslinking analysis. BSA has self-oligomerization property in accordance that oligomerized peak is shown in the negative control. The signal of oligomerized bands (o) becomes intense after photo-irradiation in 30- and 60-min. Because the average distance between **Ir-OA** and proteins is longer than that of the in-cell experiments, and there is no electron donating biomolecules, the in vitro experiment photo-crosslinking is less efficient. Condition: [**Ir-OA**] = 100 μ M, 405 nm LED light irradiation for 0, 10, 30, and 60 min (0.71 mW/cm^2), BSA loading quantity = 20 μ g. o = oligomer, t = trimer, d = dimer, m = monomer.

Fig. R4

On what data is this distribution based ?

→ Response: The distribution of **Ir-OA** and eGFPs (Figure R4) is based on protein crosslinking results and super-resolution images of Mito-eGFPs (Figure 4) and **Ir-OA** (Figure 2). Sec61b is the representative ER membrane protein, and Mito-eGFP is localized on the mitochondrial matrix by targeting sequence: MLATRVFSLVGKRAISTSVCVRAH (Fornuskova et al. *Biochem. J.*, **428**, 363-374(2010)). The information of eGFP constructs is provided in Supplementary Table 3.

Q15

“The formation of ‘protein aggregates’ include crosslinked and damaged (oxidised) proteins of the mitochondrial matrix, membrane, IMS, ER membrane, peroxisome, and maybe cytoplasm.”

“All these aggregated proteins can induce a viscous environment surrounding

mitochondria”

A complicated situation- the arguments are not very convincing.

→ Response: We are sorry for this confusing explanation. Our intention was as follows: The **Ir-OA** is located on the mitochondrial membrane (IMM and OMM) (Figure 2), so they produce ROS by photoactivation. Because the ROS is diffusive, they give oxidative damage to the proteins of the ER membrane, peroxisome, and cytoplasm as well as mitochondrial membrane. We confirmed protein modification induced by oxidative damage through protein crosslinking experiment (Figure 4) and O-Met proteomics (Figure 5). The proteins under oxidative stress could produce ‘protein aggregates’ according to previously reported papers ([i] Goosey, Zigler and Kinoshita, *Science* **208**, 1278–1280 (1980), [ii] Stadtman and Levine, *Amino Acids*. **25**, 207–218 (2003), and [iii] Tyedmers et al. *Nat. Rev. Mol.* **11**, 777–788 (2010)). Besides, previous researches have suggested that the protein aggregates can increase the viscosity of surrounding ([i] Wayne et al. *J. Phys. Chem. B*, **117**, 21, 6373–6384 (2013), [ii] Nicoud et al. *Soft Matter*, **11**, 5513 (2015), and [iii] Inthavong et al. *Soft Matter*, **15**, 4682 (2019)). Consequently, the proteins aggregates produced by photoactivation of **Ir-OA** may increase the viscosity of the surrounding local region affected by ROS. (including not only mitochondria that **Ir-OA** localizes but also ER membrane, peroxisome, and cytoplasm as contact sites).

Q17

“proteins in the cytoplasm and nucleus were partially oxidised as well,”

I note the additional data.

→ Response: We really appreciate your incisive comment.

Fig R5/Fig 2

What does the x-axis have micromolar concentration units?

Label needed for y axis

→ Response: We apologize for the severe typo. The x-axis is not concentration unit μM ; it is micrometer (μm). The missing unit of y-axis represents luminescence intensity (a.u.), thereby we revised Figure 2 as below.

Figure 2. Localisation of **Ir-OA** in living cells. A) Confocal images of **Ir-OA** with MitoTracker. Phosphorescence of **Ir-OA** (red) and fluorescence of MitoTracker (green) and merged image ($\lambda_{\text{ex}} = 405 \text{ nm}$ for **Ir-OA**, $\lambda_{\text{ex}} = 647 \text{ nm}$ for MitoTracker® Deep Red FM). Pearson's coefficient was calculated using Image J software ($R^2=0.86$). B) Airyscan confocal image of mitochondria with **Ir-OA** (red) and MitoTracker (green, outer/inner mitochondrial membrane) for identifying the specific location of **Ir-OA**. C) Airyscan confocal image of mitochondria with **Ir-OA** (red) and Mito-EGFP (green, mitochondrial matrix). Line profiling for clarifying the specific location of each emission signal was followed by corresponding images (right). The **Ir-OA** signal is well merged with the MitoTracker signal on the outer/inner mitochondrial membrane and encloses Mito-EGFP signal of the mitochondrial matrix. The mitochondria of Airyscan images were pre-swelled by photoactivation ($\lambda_{\text{ex}}=405 \text{ nm}$, 0.0125 mW) of the **Ir-OA** for identifying mitochondrial substructure (the mitochondrial swelling effect of the **Ir-OA** is explained later). Line profiling analysis was proceeded with Carl Zeiss ZEN 3.0 Blue software. Conditions: [**Ir-OA**] = $4 \mu\text{M}$, [MitoTracker® Deep Red] = 100 nM , incubation time = 2 h and 0.5 h , respectively. Source data are provided as a Source Data file.

Q32

“However, we did not need elemental analysis because high-resolution mass spectrometry (HRMS) analysis was already provided (HR-MS: m/z calculated for $\text{C}_{72}\text{H}_{66}\text{IrN}_8\text{O}_3$: $1283.4894 \text{ (M}^+)$, found: $1283.4888 \text{ (M}^+)$). The HRMS is much more accurate for identifying compounds than elemental analysis because the organic solvent or water molecules cause measurement error during the process of elemental analysis, which make it difficult to obtain consistent and accurate results”

Not true.

MS is not a good test for purity- depends on the introduction of ions into the gas phase.

Neutral molecules and non-volatile impurities are not detected.

Elemental analysis detects everything that is combustible and the presence of solvents can be accounted for (and often verified by other means)

The added HPLC is very useful but needs care with interpretation since extinction coefficients may be different for different peak (4 peaks in the present case). The detection wavelength, solvent and elution conditions are needed.

What is the HPLC purity of Ir-OC?

→ Response: In terms of purity, you are entirely right. Thus, we conducted an EA experiment for Ir-OA and Ir-OC and additional HPLC analysis for Ir-OC. As described below, the HPLC purity of Ir-OC was about 95.8% (elution condition for both of Ir-OA and Ir-OC: MeCN:H₂O = 5:5 for first 5 min and MeCN:H₂O = 2:8 for next 5 min, detection wavelength = 430 nm). Additionally, the result of elemental analysis for Ir-OA was C:60.22, H:5.24, N:7.83, O:9.81 (observed), and the calculated value is C:60.83, H:5.49, N:7.88, O:9.79 (calculated for C₇₂H₆₆IrN₈O₃Cl 5.7H₂O). For Ir-OC, C:58.72, H:4.43, N:5.78, O:5.98 (observed), and C:59.46, H:4.59, N:5.90, O:5.91 (calculated for C₄₇H₄₀IrN₄O₂Cl 1.6H₂O).

Figure R2. HPLC analysis for Ir-OA and Ir-OC. Elution condition: MeCN:H₂O = 5:5 for first 5 min and MeCN:H₂O = 2:8 for next 5 min, detection wavelength = 430 nm. HPLC purity was 96.5% for Ir-OA and 95.8% for Ir-OC.

Accordingly, a sentence was added to the supporting information as below.

Page 6 Line 243

'HPLC purity: 95.8%. Elemental analysis: calculated for C₄₇H₄₀IrN₄O₂Cl 1.6H₂O = C:59.46, H:4.59, N:5.90, O:5.91, found: C:58.72, H:4.43, N:5.78, O:5.98'

Page 7 Line 265

'Elemental analysis: calculated for C₇₂H₆₆IrN₈O₃Cl 5.7H₂O = C:60.83, H:5.49, N:7.88, O:9.79, found: C:60.22, H:5.24, N:7.83, O:9.81.'

Q34

Noted that "we removed MTT assay experiments with A549 cells because this experiment was not logically connected to the rest parts of the manuscript"

→ Response: We really appreciate your incisive comment.

Other comments

Fig 1A: why has a specific enantiomer been drawn? A comment needs to be added on the chirality of the complexes they have used (presumably racemic).

→ Response: Thank you for your constructive comment. As you pointed out, the synthesized **Ir-OA** is probably racemic, and the other enantiomers can exist. Therefore, we added a sentence to the legend for Figure 1A as below.

Page 21 Line 532

“The illustrated **Ir-OA** molecule is a Δ isomer, but the Λ form enantiomer can exist.” (added)

Fig 1C: absorbance does not have arbitrary units (a.u.). In fact it does not have units at all.

(log to base 10 of incident/transmitted radiant flux)

→ Response: It is totally true. We removed the ‘(a.u.)’ mark of Figure 1C. Thank you for your incisive comment.

Figure 1. Photophysical characterisation of iridium(III) photosensitiser. A) Schematic illustration of the molecular engineering strategy: intramolecular energy transfer and resulting photoactivated ROS generation. The illustrated **Ir-OA** molecule is a Δ isomer, but the Λ form enantiomer can exist. B) Molecular structure of **Ir-OA**, **Ir-OC**, and compound 4. C) UV-vis absorption spectrum of the three presented chemicals. D) Subsequent emission spectra ($\lambda_{ex} = 400$ nm) of the three chemicals. Magnified emission spectra show the enhanced emission of **Ir-OA** compared with that of **Ir-OC**, which reveals the evidence of energy transfer.

Conditions for the absorption and emission spectra; [**Ir-OA** or **Ir-OC** or compound 4] = 20 μM in $\text{H}_2\text{O}:\text{DMSO} = 99:1$ (v/v%). E) $^1\text{O}_2$ assay using the absorbance decay of 9,10-anthracenediyl-bis(methylene)dimalonic acid (ABDA) under light exposure. The ABDA is degraded by $^1\text{O}_2$ produced by photoactivation of the **Ir-OA**. F) $\text{O}_2^{\bullet-}$ assay using the fluorescence enhancement of dihydrorhodamine 123 (DHR123). DHR123 is oxidised to rhodamine123 by the produced $\text{O}_2^{\bullet-}$, which enhances fluorescence signal. Conditions for ROS assays: [iridium(III) complexes] = 4 μM ; [ABDA] = 100 μM or [DHR123] = 4 μM in $\text{H}_2\text{O}:\text{DMSO} = 999:1$ (v/v%). All error bars = s.d. (n=3). Source data are provided as a Source Data file.

Reviewer #2 (Remarks to the Author):

The authors have thoroughly addressed all the reviewer comments, and the manuscript that they have presented is much clearer, and likely to be very impactful.

Two minor points:

Page 3, line 43 – “Oxidative stress induced by photosensitisers causes chemical modifications of biomolecules, which include proteins, unsaturated lipids and DNA.” Is confusing as it suggests that proteins, unsaturated lipids and DNA are types of biomolecules (which is, of course, true, but not the intended point. “Oxidative stress induced by photosensitisers causes chemical modifications of biomolecules including proteins, unsaturated lipids and DNA” or “Oxidative stress induced by photosensitisers causes chemical modifications of biomolecules such as proteins, unsaturated lipids and DNA” would be much clearer

Page 5, line 64 – “The proposed mechanism will aid the understanding of cancer therapeutics induced mitochondrial oxidative stress.” Should be “The proposed mechanism will aid the understanding of how some cancer therapeutics can induce mitochondrial oxidative stress.”

→ Response: We revised the two sentences as you pointed out. We appreciate for your careful comments.

Page 3 Line 44

Oxidative stress induced by photosensitisers causes chemical modifications of biomolecules, which include proteins, unsaturated lipids, and DNA. → Oxidative stress induced by photosensitisers causes chemical modifications of biomolecules including proteins, unsaturated lipids, and DNA.

Page 4 Line 64

The proposed mechanism will aid the understanding of cancer therapeutics induced mitochondrial oxidative stress. → The proposed mechanism may aid the understanding of how some cancer therapeutics can induce mitochondrial oxidative stress.

Reviewer #3 (Remarks to the Author):

I believe that the authors have addressed all the concerns that i raised during the initial review and I am happy to recommend publication of this manuscript in your journal.

REVIEWERS' COMMENTS

Reviewer #2 (Remarks to the Author):

The authors have thoroughly addressed all outstanding reviewer comments, and this work is appropriate for publication.